# FDG-PET/CT in the Monitoring of Lymphoma Immunotherapy Response: Current Status and Future Prospects

**DOI:** 10.3390/cancers15041063

**Published:** 2023-02-07

**Authors:** Akram Al-Ibraheem, Ahmed Saad Abdlkadir, Malik E. Juweid, Kamal Al-Rabi, Mohammad Ma’koseh, Hikmat Abdel-Razeq, Asem Mansour

**Affiliations:** 1Department of Nuclear Medicine and PET/CT, King Hussein Cancer Center, Al-Jubeiha, Amman 11941, Jordan; 2Department of Radiology and Nuclear Medicine, Division of Nuclear Medicine, University of Jordan, Amman 11942, Jordan; 3Department of Medical Oncology, King Hussein Cancer Center, Amman 11941, Jordan; 4Department of Internal Medicine, King Hussein Cancer Center, Amman 11941, Jordan; 5Department of Internal Medicine, School of Medicine, University of Jordan, Amman 11942, Jordan; 6Department of Diagnostic Radiology, King Hussein Cancer Center, Amman 11941, Jordan

**Keywords:** immunotherapy 1, immunotherapy in lymphoma 2, FDG PET/CT 3, metabolic PET parameters 4, lymphoma immunotherapy response criteria 5, immuno-PET 6

## Abstract

**Simple Summary:**

Cancer immunotherapy is a type of cancer treatment that uses the immune system to fight cancer cells. Some of these treatments stimulate the immune system, while others prime the immune system to identify better and target cancer cells. In parallel with the implementation of cancer immunotherapy, therapy-specific FDG PET/CT response criteria were explicitly designed specifically for that purpose. The goal of this review article is to discuss the effects and side effects of cancer immunotherapy and to correlate this with the proposed criteria and relevant patterns of FDG PET/CT in lymphoma immunotherapy as applicable. Additionally, the latest updates and future prospects will be explored.

**Abstract:**

Cancer immunotherapy has been extensively investigated in lymphoma over the last three decades. This new treatment modality is now established as a way to manage and maintain several stages and subtypes of lymphoma. The establishment of this novel therapy has necessitated the development of new imaging response criteria to evaluate and follow up with cancer patients. Several FDG PET/CT-based response criteria have emerged to address and encompass the various most commonly observed response patterns. Many of the proposed response criteria are currently being used to evaluate and predict responses. The purpose of this review is to address the efficacy and side effects of cancer immunotherapy and to correlate this with the proposed criteria and relevant patterns of FDG PET/CT in lymphoma immunotherapy as applicable. The latest updates and future prospects in lymphoma immunotherapy, as well as PET/CT potentials, will be discussed.

## 1. Introduction

Lymphoma immunotherapy is becoming more appealing over time, as evidenced by the wide variety of approved therapy options that are now available for certain stages and subtypes of lymphoma [1,2,3]. Applying this therapy in clinical practice has broadened the concept of lymphoma treatment, as it can fight tumor biology in addition to on-site disease eradication [4]. Management of lymphoma has seen great advances in recent years, with a shift in focus to immunotherapy in the last three decades [5]. This has translated to FDA approval of several immunotherapies. Positron emission tomography coupled with computed tomography (PET/CT) allows for better evaluation of response to immunotherapy in FDG-avid lymphomas, as well as providing prognostication insights [6]. Metabolic PET parameters are reliable predictors in the context of absent alternative biomarkers.

This paper will explore cancer immunotherapy with a focus on lymphoma. Specifically, it will discuss the role of FDG PET/CT in the immunotherapy of FDG avid lymphoma. Finally, it will address patterns of immunotherapy-related toxicities and future directions.

## 2. Cancer Immunotherapy: Classification, Previous and Current Facts

### 2.1. Historical Overview of Cancer Immunotherapy

The concept of cancer immunotherapy has been explored and studied since the 19th century [7]. However, its clinical implementation remained debatable until the approval of the first immunotherapy drug in 1976 (Figure 1). The first generation of immunotherapy relies on the action of vaccines to boost the immune response. This was followed by the utilization of anti-tumor cytokines, monoclonal antibodies, oncolytic viruses, and adoptive cell therapies to recruit immune cells against certain types of cancers. Thus far, there are many types of cancer immunotherapies that are implemented in lymphoma treatment.

### 2.2. Immunotherapy in Lymphoma

#### 2.2.1. Previous Footprints

After years of research and experimentation, it became evident in the last century that certain bacterial vaccines, such as Bacille Calmette-Guérin (BCG), could recruit immune cells to inhibit the recurrence of urinary bladder cancer [4,8]. This approach remains active and is still adopted in clinical practice. This progress led to the adoption of the cytokine family later on. Cytokines were found to be effective in inhibiting tumor cell proliferation and enhancing cancer apoptosis [9]. The timeframe between interferon discovery and adoption witnessed the discovery of interleukins (IL), namely IL-2. IL-2 was found to be effective in treating advanced renal cell carcinoma (RCC) and metastatic melanoma [10,11]. 

With the aim to reinforce passive immunotherapy, researchers are investigating new ways to harness the power of the immune system to fight cancer. One promising area of research is using viruses to target cancer cells and boost the immune system’s response to the tumor environment [12]. This approach has shown success in treating melanoma with genetically modified herpes viruses [13,14].

#### 2.2.2. Monoclonal Antibodies: From Rituximab to Immune Checkpoint Inhibitors

Immunomodulating antibodies were studied extensively in the 1990s, and rituximab emerged as a prototype for many other monoclonal antibodies in 1997 [8,15]. It is incorporated with cyclophosphamide, doxorubicin HCl, vincristine, and prednisone in the R-CHOP protocol for the treatment of diffuse large B-cell lymphoma (DLBCL) [16,17,18,19]. The adoption of the immunochemotherapy protocol (R-CHOP) has been shown to have a much higher survival impact compared to standard chemotherapy protocols [20].

Under the same umbrella, immune checkpoint inhibitors (ICI) became approved and available for many cancer types. The FDA approved the use of ipilimumab in 2011 as a therapy for advanced melanoma [21,22]. In 2016, Nivolumab attained the same approval as the first programmed cell death protein 1 (PD-1) inhibitor for the treatment of Hodgkin’s lymphoma (HL) [23]. Within the following year, another anti-PD-1 inhibitor (pembrolizumab) was approved. The current range of ICI drugs use checkpoint blockade as their primary mode of action, yet there are distinctions between the various subclasses. To date, only a limited number of ICIs have been approved for clinical use by the FDA (Table 1), with others likely to follow in the foreseeable future. The ICI has been shown to be effective in clinical settings against HL cells and the tumor microenvironment [24]. The programmed death ligand-1 programmed cell death protein (PD-L1) is potentially blocked and inhibited through ICI administration, which was observed in 70% of cases [25]. This inhibitory pathway can terminate tumor growth and stimulate the immune system against HL cells [26,27].

More recently, FDA approval of Brentuximab Vedotin (BV)was attained in 2018. Approval was gained after successful results obtained from a randomized clinical trial of 1334 patients that have received either BV plus doxorubicin, vinblastine, and dacarbazine (BV-AVD) or bleomycin plus AVD (ABVD) [28]. At first, BV was granted approval to treat advanced adult HL, followed by recent FDA approval of pediatric untreated classical HL. 

#### 2.2.3. Adoptive Cell Therapy

In recent years, researchers have become more interested in targeting both genetic and cellular abnormalities in tumors in order to better control cancer growth and spread [29]. A new type of T cell therapy, known as chimeric antigen receptor therapy (CAR-T), has emerged as a promising treatment option [30]. In this therapy, T cells are taken from a patient’s blood and modified to express artificial receptors specifically targeted at a particular tumor antigen [31]. This allows the T cells to bind to and kill the cancer cells while leaving healthy cells unharmed. In detail, the patient’s T-cell will be equipped with an artificial CAR. These receptors are composed of an antibody-derived single-chain variable fragment, a transmembrane, and a signaling domain [32]. The CAR segment will allow T-cells to target tumor antigens. Through antigen binding, the CAR will induce cytokines recruitment and proliferation against receptor-specific cancer cells [33].

CAR-T cells were approved for use in leukemia in 2017 and then for lymphoma in the following year [34,35,36]. CAR-T cells represent a major advancement in the field of immunotherapies. This is evident from the fact that two CAR-T cells, Yescarta (axicabtagene ciloleucel) and Kymriah (tisagenlecleucel), are commercially available [6].

#### 2.2.4. Bispecific Antibodies: The Most Recent Addition to the Group

Bispecific antibodies, also known as bispecific T-cell engagers (BiTes), are novel protein constructs that target both B-cells (CD20) and T-cells (CD3) [37]. These include Mosunetuzumab, Glofitamab and Epcoritamab. Mosunetuzumab is already FDA/EMA approved for use in r/r Follicular lymphoma (FL). The remainder of BiTes are on the way to being granted the same approval. 

## 3. Role of PET/CT in Lymphoma Immunotherapy

The lack of reliable biomarkers to measure immunotherapy response has made FDG PET/CT response criteria more useful. This hybrid imaging modality can use various metabolic parameters to predict and evaluate therapy response. In fact, FDG PET/CT is the only imaging modality with the ability to evaluate therapy response and demonstrate metabolic aspects of immunotherapy-related side effects [38].

### 3.1. Response Patterns

The response to immunotherapy varies among patients, with most experiencing a good initial response [27]. However, some patients may demonstrate different patterns of response after therapy administration (Figure 2).

#### 3.1.1. Pseudoprogression

Some patients may initially appear to experience disease progression (known as pseudoprogression) before ultimately achieving a favorable clinical picture [39]. This false positive pattern was first reported in 15% of patients receiving anti-CTLA4 therapy [39]. The need to correct initial erroneous positive results necessitates the implementation of new response criteria [40,41]. Pseudoprogression was first observed in solid tumors and later reported in lymphomas, introducing additional confusion in PET-driven response assessment [27,42,43]. Rather than being indicative of actual progression, pseudoprogression is more similar to a flare phenomenon caused by massive immune stimulation (Figure 3) [27]. It is also possible that a delayed immunologic response may contribute to pseudoprogression [44]. The maximum increase in tumor burden linked with pseudoprogression has been reported to range from 20% to 163% [45]. During the initial phases of therapy, immune cells may be recruited into the tumor microenvironment, leading to a temporary increase in tumor size and metabolic activity [27]. However, pseudoprogression can be confirmed during follow-up imaging through eventual tumor regression and favorable clinical outcomes [46]. Nonetheless, it remains important to weigh the possibility of initial radiographic progression being pseudoprogression against the potential for true progression. A biopsy of enlarged lesions for pathologic review may be helpful in identifying pseudoprogression [47]. McGehee et al. reported a case of pseudoprogression in a T-cell lymphoma patient [48]. In addition to being symptom-free, the biopsy for the patient was negative, ruling out disease progression [48]. The same findings were previously mentioned in a case series [49].

Although biopsy of enlarged lesions can be helpful in distinguishing pseudoprogression from true disease progression, it is not always practical for some patients (Figure 4). For example, asymptomatic patients usually tend to express this pattern later on (Figure 5). Therefore, clinical correlation and careful follow-up are the keys to better tracing the disease’s progress. Pseudoprogression is usually accompanied by an initial increase in tumor size but not by a deterioration in the patient’s clinical condition or performance status. Genuine disease progression, or hyperprogression, on the other hand, is often accompanied by a deterioration in the patient’s condition.

#### 3.1.2. Hyperprogression

More recently, a permanent progression has been observed and evidenced by the increased rate of tumor growth [50,51]. This was supported by Phase 3 clinical trials, which demonstrated decreased survival outcomes in some patients who underwent immunotherapy treatment [50,51]. Under the category of hyperprogression, this pattern most commonly affects elderly patients and has been noted in up to 29% of patients receiving immunotherapy [52,53]. In hyperprogression, there is evidence of a dramatic tumor growth rate associated with clinical worsening [50]. When comparing baseline imaging with initial therapy imaging, there is a minimum of a twofold increase in overall tumor burden (Figure 6). In such cases, the only choice is to terminate immunotherapy [50,51].

#### 3.1.3. Potentiating Abscopal Effect

The findings suggest that combining radiotherapy with immunotherapy may boost the abscopal effect of local radiotherapy treatment (Figure 7) [27]. This response pattern was first observed in the 1950s after researchers noted clinical responses at distant metastatic sites following the administration of locoregional radiotherapy [54]. Later research showed that this phenomenon is mediated by T cells and that the incidence of the abscopal effect is favorable in immunocompetent patients [55]. Enhancing immune system response through immunotherapies can therefore result in a potential synergistic effect [56,57]. Researchers are still working to determine the exact mechanism of this effect after several reported clinical cases [56,57].

### 3.2. PET Response Criteria in Lymphoma

Given the lack of biological markers to assess the efficacy of immunotherapy [58]. It was necessary to create therapy-specific criteria to assess the wide array of response patterns encountered.

#### 3.2.1. Lugano Classification

The Lugano criteria are widely used in studies and clinical trials of immunotherapy drugs, despite being non-specific for immunotherapy response [59,60,61]. The criteria provide a solid foundation for future therapy-specific response criteria [46,62]. In 2014, the Lugano classification was adopted by a team of specialists in oncology, hematology, radiology, and nuclear medicine [26]. This classification uses metabolic PET parameters to assess response to therapy at the end of therapy (EoT) and during the interim period (iPET) [63]. Since then, it has been considered the gold standard interpretation criteria for FDG-avid lymphomas [63,64]. The Lugano study introduced a five-point scale for assessing metabolic response instead of using a dichotomous response pattern [65]. The ordinal scale, consisting of five Deauville scores (DS) (Table 2), was used to examine the degree of response [26]. The degree of response can be measured by qualitative visual assessment of FDG uptake within the most intense residual lymphomatous lesion identified during EoT or i-PET [26]. The retrieved values are then visually compared to reference metabolic values derived from the background, mediastinal blood pool, and liver. The complete metabolic response is indicated if the Deauville 5-point scale (5PS) is between 1–3. A score of 4 or 5 on the five-point scale, with evidently reduced uptake since the baseline study, is indicative of partial metabolic response (PMR) [63]. A DS of 4 or 5 can also indicate disease progression (PD) when there is evidence of increased uptake since the baseline study, with or without the interval development of new FDG avid lymphomatous disease process [63].

#### 3.2.2. Lymphoma Response to Immunomodulatory Therapy Criteria (LYRIC)

The introduction of immunotherapy has necessitated modifications to the previously established Lugano classification in order to account for pseudoprogression [66]. This transient pattern of response, characterized by an increase in tumor size and metabolic flare, can make it difficult to determine if a patient is truly progressing or if their disease is simply pseudo-progressing [67]. In 2016, the LYRIC group introduced a new category of response, labeled the Indeterminate Response (IR), to account for this pseudoprogression pattern [65]. If there is suspicion of pseudoprogression, a patient can be classified as IR, and therapy can be continued for up to 12 weeks before a definitive confirmation is made(Appendix A) [27]. After that point, follow-up FDG PET/CT imaging can be used to discriminate between true progression and pseudoprogression. Additionally, histopathologic confirmation can be pursued to better understand the disease pattern [46].

#### 3.2.3. Response Evaluation Criteria in Lymphoma (RECIL)

In an effort to harmonize lymphoma response criteria in clinical trials, RECIL revolutionized the way to evaluate immunotherapy [62]. Anatomically, lesions measurement was modified to include only unidimensional measurement of the long diameter of three selected target lesions [62]. In addition, RECIL proposed measuring and comparing the difference in tumor burden. As a result, complete response (CR) has replaced CMR and would require at least a 30% reduction in tumor burden in addition to a DS range of 1–3 [62]. In a parallel fashion, a higher DS of 4–5 would indicate a PR, achieving a reduction in tumor burden of at least 30% [62]. A new category labeled as a minor response has been proposed in cases of at least 10% tumor burden reduction, not exceeding the 30% threshold [62]. A stable disease pattern can be observed if the range of change in the tumor burden lies between −10% and +20%. Otherwise, disease progression implies a value of more than a 20% increase in tumor burden, with or without the appearance of a new lesion [62]. It is noteworthy that both the PD and minor response categories do not require correlation with 5PS [62]. Additionally, disease relapse is considered when a newly appearing lesion exceeds 1.9 cm in the long axis [27,62].

### 3.3. FDG PET/CT in Hodgkin’s Lymphoma (HL)

HL is known for its high FDG avidity and chemosensitivity [68,69,70]. Previous research has focused on exploring standard therapeutic protocols to determine therapy response and outcome [68,69,70,71]. However, with the emergence of new lines of cancer immunotherapy, new treatment protocols have been introduced, accompanied by new PET/CT response criteria specific to lymphoma immunotherapy [62,65].

#### 3.3.1. Immune Checkpoint Inhibitors 

The idea behind the ICI mechanism has led to the first clinical trial conducted by Ansell et al. [72]. This first clinical trial reported a response rate of nearly 90% in relapsed or refractory (R/R) HL cases treated with nivolumab (Appendix A) [72]. In a similar fashion, the efficacy of pembrolizumab in BV-relapsed HL was examined, resulting in an overall response rate (ORR) of about 65% [73]. In the last decade, a few other clinical trials and studies have been conducted to evaluate the safety and efficacy of anti-PD1 agents, such as nivolumab and pembrolizumab for R/R HL, which have led to FDA approval of these agents for this use [74,75]. In one of the earliest studies exploring the role of FDG PET/CT in anti-PD1 immunotherapy, Dercle et al. found that among 16 patients with HL, 4 achieved complete metabolic response after 6 months of therapy [61]. The same patients were then analyzed to determine patterns of response, and it was found that 9 out of 16 patients had objective response [76]. These findings confirmed the reliability of FDG PET/CT in assessing response to immunotherapy [76]. Metabolic PET parameters were also found to be significant outcome predictors. A study by Castello et al. found that metabolic PET parameters are significant outcome predictors in patients with R/R HL treated with pembrolizumab [59]. A total of 43 patients were enrolled, and after a median follow-up of 19 months, the responder group had lower DS and SUV max values when compared with the non-responder group [59]. Additionally, progression free survival (PFS) was found to be longer in the responder group [59]. Despite the previously stated achievements of ICI in clinical settings, long-term benefits are still not adequately studied [77]. Many recent studies have explored the potential benefits of combining immunotherapy with other treatment protocols for patients with advanced HL [78,79,80]. One such study found that the combination of nivolumab with BV vedotin plus doxorubicin and dacarbazine was highly effective, with more than 90% of patients achieving an ORR of 93% [81]. A recent multicentric study incorporating two novel agents including Nivolumab and BV explored disease response following 3 cycles of BV-AVD at an interim period in limited HL patients [82]. Patients with iPET negative results received consolidation therapy with nivolumab, while iPET-positive patients received four cycles of Nivolumab plus BV followed by Nivolumab Consolidation [82]. All iPET negative patients achieved CR after assessment by PET/CT at EoT [82]. iPET positive patients was found only in two patients, one of which achieved CR at EoT while the other one has DS of 4 achieving CR after switching to radiotherapy due to treatment side effects [82]. These findings suggest that the combination of different treatment protocols may be a promising approach for improving outcomes in HL.

#### 3.3.2. Brentuximab Vedotin (BV)

After the approval of immune checkpoint inhibitors for HL, another monoclonal antibody became available, BV, for the treatment of both adult and pediatric patients. The efficacy of the drug was explored in many trials through the use of FDG PET scans for therapy response [83,84,85]. These studies showed an ORR of around 70%. Additionally, Kahraman et al. examined the efficacy of BV in clinical settings through the use of FDG PET/CT scans to monitor therapy outcomes in cases of R/R HL [86]. At the interim period and after a median follow-up of 16 months, PFS was significantly prolonged in patients with negative interim PET results compared to positive interim results [86]. A recent study confirmed the previous observation that patients with negative iPET results have improved PFS and overall survival (OS) compared to those with positive iPET results [87]. 

BV was also incorporated as frontline therapy substituting Bleomycin in BV-AVD regimen. This treatment protocol was tested in phase III ECHELON 1 study [88]. The main aim was to compare standard ABVD treatment vs. BV-AVD combination for patients with advanced HL [88]. The trial included six cycles of treatment, and an iPET was performed after cycle 2 [88]. A modified PFS was implemented counting any event of additional anticancer therapy as part of progression in patients exceeding 5PS of 3 [88]. The modified 2-year PFS for those receiving BV-AVD was 82.1%, while the PFS for those receiving ABVD was 77.2% (*p* = 0.04) [88]. A recent post hoc analysis has found that patients with stage IV disease and extranodal sites seem to benefit the most from BV-AVD in terms of modified PFS [28]. It’s noteworthy that the modified PFS benefit with BV-AVD was largely limited to patients who had a positive iPET (DS of 4–5) after 2 cycles of therapy (57.5% vs. 42.0%; HR, 0.61; 95% CI, 0.34–1.09 vs. 85.2% vs. 80.9%; HR, 0.77; 95% CI, 0.59–1.02 for those who had a negative iPET) [89]. Similarly, standard PFS was calculated for north American subgroup and was found to be higher in BV-AVD arm (88.1% vs. 76.4%; HR, 0.50; 95% CI, 0.32–0.79; *p* = 0.002) [90]. The ECHELON-1 trial results suggest that adding BV to AVD regimen may increase efficacy of initial therapy, for patients with advanced HL.

In an effort to minimize toxicity at escalation, The German Hodgkin Study Group has tested variations of the regimen in which BV is added to frontline therapy [91]. These regimens do not contain bleomycin or vincristine in an effort to prevent the worsening of pulmonary toxicity or neuropathy. Procarbazine was changed to dacarbazine in one variant to lessen the risk of secondary leukemia [91]. 104 patients with stages IIB to IV HL participated in a phase 2 trial where they were randomly assigned to receive either 6 cycles of brentuximab vedotin, etoposide, cyclophosphamide, adriamycin, procarbazine, and prednisone (BrECAPP) or 6 cycles of brentuximab vedotin, etoposide, cyclophospham (BrECADD) [91]. The 18-month PFS estimates were 95% and 89% for patients receiving BrECAPP and BrECADD, respectively, with a median follow-up of 17 months [91]. 

Advanced HL cases were assessed with PET derived metrics to determine the predictive value. A study by Gavane et al. included 45 patients with R/R HL treated with BV-based salvage therapy [92]. It was observed that several baseline metabolic PET parameters, including metabolic tumor volume (MTV), total lesion glycolysis (TLG) and SUV peak, provide significant prognostic value in such patients [92]. Previous research has established the role of BV in advanced cases of HL, but its efficacy in early-stage HL is not as well-known. In a study by Abramson et al., the use of combined AVD/BV without radiotherapy in 34 patients with non-bulky early-stage HL was explored [93]. One cycle of BV was administered on days 1 and 15, followed by four cycles of AVD/BV [93]. A complete response rate of 52% after the lead-in cycle of BV and 97% after two AVD/BV cycles was achieved, and the 3-year PFS rate was 94% [93]. In a study by Park and colleagues, the approach of 6 cycles BV consolidation therapy after 2–6 ABVD cycles in early-stage HL was explored [94]. A consolidation approach yielded a 95% complete response rate, and a three-year progression-free survival of 92% [94]. Currently, there is a noticeable shift in emphasis toward incorporating BV as part of frontline therapy to observe therapy outcomes.

#### 3.3.3. Chimeric Antigen Receptor Therapy (CAR-T)

Interestingly, CAR-T is the only immunotherapy that would require FDG PET/CT assessment during initial administration. In fact, two FDG PET/CT studies have to be carried out before CAR-T infusion. This involves performing FDG PET/CT at time of decision (TD), followed by second scan at time of transfusion (TT). Afterwards, another two scans will be performed to monitor therapy response at 1-month (M1) and 3-month (M3) intervals [95]. This approach has attained high sensitivity and specificity of about 99% and 100% respectively [96,97]. It is noteworthy that not all clinical centers adhere to this approach, as many clinicians rely on TT PET/CT as a baseline study [95]. When assessing treatment response in clinical settings, multiple PET parameters are usually incorporated. These parameters are derived from values of 5 PS, SUVmax, and the variation between different time points (∆SUVmax), along with tumor volume analyses [95]. Volumetric analyses rely on values of MTV. An unfavorable response is considered when there is less than 66% of SUVmax reduction between two time points [98,99,100]. The successful results of BV therapy have helped to provide the infrastructure for CAR-T to implement CD30 as a potential target. In a study of 18 patients with R/R HL, Wang and colleagues found that treatment with anti-CD30 CAR T cells was feasible and tolerable [101]. Patients in the study had received extensive prior treatment, including both conventional lymphodepletion regimens and more disease-controlling regimens [101]. The ORR in the study was 39%, with 28% of patients showing stable disease at two months after therapy infusion. The median PFS was 6 months [101]. Further support for the safety and efficacy of anti-CD30 CAR-T therapy comes from a phase 1 trial conducted by Ramos et al. [102]. This trial included 9 patients with R/R HL or anaplastic large cell lymphoma. The study showed an ORR of 33%, demonstrating the feasibility and tolerability of this type of therapy [102]. A more recent trial enrolling 41 patients with R/R HL showed even more promising results, with an ORR of 72% and a one-year overall survival rate of 94% [103]. This study suggests that anti-CD30 CAR-T therapy is a promising treatment option for patients with R/R HL [103]. In a similar fashion, Voorhees et al. examined the predictive role of MTV prior to anti-CD30 therapy in HL [104]. This study found that there was a strong association between PFS and MTV prior to lympho-depletion [104]. Therefore, minimizing MTV value before CAR-T is found beneficial. The results from this study have broadened the field of research to include CAR-T. To date, there are 4 clinical trials underway to explore different potential uses of PET/CT for assessing therapy response in CAR-T patients [105,106,107,108].

### 3.4. FDG PET/CT in Non-Hodgkin’s Lymphoma (NHL)

Similarly, FDG PET/CT is of vital importance for outcome prediction and prognostication [101]. The only difference in NHL is that there are certain histologic subtypes that do not optimally express FDG avidity [109]. Indolent NHL fall under the category of variable FDG avid NHL while aggressive NHL usually have moderate to high FDG avidity (Table 3). Therefore, incorporation of FDG PET/CT is most acknowledged in response assessment of aggressive NHL.

### 3.5. FDG PET/CT in Diffuse Large B-Cell (DLBCL)

#### 3.5.1. Rituximab

Since the approval of FDG PET/CT by the FDA, a number of studies have been conducted to explore the efficacy of this treatment modality. Haioun et al. were among the first to examine the prognostic and predictive value of early FDG PET/CT imaging [110]. In their study, 41% of all 90 patients received rituximab as part of the therapy protocol [110]. These patients were then followed up to determine the prognostic outcome [110]. It was concluded that patients with negative PET results had much more favorable outcomes, reflected by PFS and OS rates of 82% and 90%, respectively, as compared to 43% and 61% for those with positive PET results [110]. More recently, a group of DLBCL patients treated with R-CHOP and evaluated by FDG PET/CT at the interim stage were prospectively enrolled in a study [111]. The calculated 3-year PFS and OS rates in iPET negative patients achieved statistically significant superiority when compared to positive results [111]. Data from these studies along with others were collected to conduct a meta-analysis [112]. This meta-analysis was interested in examining the predictive role of iPET in DLBCL patients treated with R-CHOP [112]. The overall sensitivity and specificity of iPET were observed to be discouraging, justifying the need for more effort to unify response criteria [112]. Recently, a group of GOYA DLBCL patients were analyzed for data following the first line of immunochemotherapy to determine survival outcome [113]. It was found that EoT PET is an independent predictor of both PFS and OS and a promising prognostic marker for such patients [113]. The results from the previous analysis necessitate a meta-analytic study in the near future to support this evidence. Another GOYA group analysis was carried out to observe PFS rates difference between DLBCL patients receiving R-CHOP vs. Obinutuzumab plus CHOP (G-CHOP) [114]. The study was unable to demonstrate any PFS benefit of G-CHOP over R-CHOP in previously untreated patients with DLBCL [114]. [111]As of right now, metabolic PET parameters are being extensively studied in an attempt to outline their prognostic values [115,116,117]. The GOYA group of patients was analyzed using metabolic PET parameters. Tumor MTV was found to be a predictor of therapy failure in these patients. A recent study concerning baseline PET parameters in R-CHOP treated DLBCL patients was conducted [115]. Metabolic PET parameters were used including SUVmax, SUVmean, MTV and TLG [115]. The study suggests that these parameters may have a prognostic value at baseline and interim intervals [115]. In another study, tumor MTV values were found to be the most reliable parameters among all to determine survival outcome [116]. This was recently shown by another study that confirmed the predictive value of baseline and interim MTV on survival outcome [117]. It appears obvious by now that metabolic PET parameters can establish solid background for future response-adapted management approaches in NHL patients.

#### 3.5.2. Immune Checkpoint Inhibitors (ICI)

Unlike HL, the results achieved in the previous literature using ICI in NHL are less encouraging. Despite having high safety profile, ICI failed to achieve optimal efficacy in NHL. The safety and efficacy of nivolumab in DLBCL were assessed in the previous single-arm phase II study by Ansell et al. [118]. The study acknowledged suboptimal ORR despite the highly observed safety profile [118]. On the other hand, Results from clinical trials of ICI combined with other immunochemotherapies appears more promising. Pembrolizumab was explored as a treatment for DLBCL in a study of 30 patients [119]. This study found that the combination of pembrolizumab and R-CHOP resulted in an ORR of 90%, a CR of 77%, and a 2-year PFS of 83% [119]. The findings of this trial indicate that combining the PD-L1 inhibitor atezolizumab with chemotherapy may be a promising treatment option for DLBCL. The combination of atezolizumab and R-CHOP (a type of chemotherapy) resulted in a high efficacy, with an ORR of 87.5% and durable responses in 80% of patients at 24 months [120]. Based on previous research, it appears that combining immunotherapy with chemotherapy is more likely to result in favorable outcomes in terms of response and clinical outcome.

#### 3.5.3. Chimeric Antigen Receptor Therapy (CAR-T)

A few studies have examined the value of FDG PET/CT in CAR-T, with mixed results. Shah et al. were among the first to examine MTV in a small group of NHL patients and found that nearly half the patients had non-measurable MTV values at M1 (Appendix A) [121]. These patients showed long-term remission over the following 2 years [121]. The other half presented with measurable MTV and witnessed an early relapse [121]. Cohen et al. have approached the issue differently and include both DS and ∆SUVmax for evaluation [122]. It was concluded that SUVmax prior to therapy may help determine treatment eligibility and that DS and ∆SUVmax can help identify treatment failure [122]. Recently, Galtier et al. conducted a multicentric cohort study which highlighted the high predictive values of both the 5 PS and MTV [123]. This was also previously explored by Kuhnl et al., who found that Deauville criteria may predict the risk for CAR-T failure and help direct post-CAR-T management [124]. Breen et al. have conducted more detailed analysis of SUVmax values at M1 and found that higher SUVmax values indicate higher risk for disease progression [125]. SUVmax above 10 at M1 is regarded as a significant prognostic and predictive indicator in patients with stable disease or partial response [125]. This was later confirmed by Al Zaki et al. [126].In NHL, tumor burden was validated through the use of FDG PET/CT in a retrospective study by Wang et al. [127].In fact, having high tumor burden at baseline was linked to more aggressive cytokine release syndrome [127]. Bailly et al. enrolled a group of R/R NHL patients in order to demonstrate the added value of adequate disease control prior to therapy [128]. Among all 40 patients, 33 cases were adequately managed prior to CAR-T [128]. During TT PET/CT, adequately treated patients showed more favorable outcomes in terms of event free survival when compared to others [128]. Moreover, 5 of the remainder 7 patients have witnessed early disease relapse [128]. Therefore, adequate control prior to CAR-T was linked to more favorable response in such cases. Despite the encouraging outcomes of previous studies, more research is needed with larger cohorts to get a complete picture.

### 3.6. FDG PET/CT in Follicular Lymphoma (FL)

Follicular lymphoma (FL) is one of the most common types of lymphoma, representing 22% of adult non-Hodgkin’s lymphomas (NHL) worldwide [129]. The disease can present with a variable clinical course, usually indolent and slow growing, while in other cases the disease may become aggressive, often characterized by histological transformation into a high-grade lymphoma (25–60%) and early death [130]. FL belongs to a group of neoplasms usually presenting with a variable FDG avidity. Therefore, permitting an overall good diagnostic accuracy using FDG PET/CT, up to 98% [131]. Although the outlook for patients with FL has improved in recent years, with a median survival that can exceed 20 years, FL is still considered incurable [132]. The main goal of treatment is usually disease control and extending patients’ life expectancy [133]. 

#### 3.6.1. Rituximab

In FL, the combination of rituximab and chemotherapy has been shown to improve outcomes for patients with FL. However, 20% of patients treated with this immunochemotherapy still experience disease progression within a short time frame, and 50% of them will witness death within 5 years [134]. FDG PET/CT have quickly replaced CI through the use of metabolic PET parameters. Providing more reliable indices for therapy response and outcome. In result, staging, therapy response and surveillance became more accurate. In 2011, a study by Trotman et al. was the first to provide large-scale evidence that EoT PET/CT after Immunochemotherapy treatment is a strong and independent predictor of PFS in FL [135]. This study included 160 patients from the prospective Primary Rituximab and Maintenance (PRIMA) study group [135]. Disease progression and death was significantly higher in PET-positive patients (70.7% at 42 months) compared to PET-negative patients [135]. The study also showed that the predictive value of the FDG PET/CT is independent of the state of the response by CI [135]. In result, FDG PET/CT can function as metabolic biomarker to viable disease process. Dupuis et al. have also examined prognostic role of FDG PET/CT at both interim and EoT periods [136]. This study included a total of 121 FL patients with median follow-up of 23 months. Among all patients, 116 cases have received at least 4 cycles of R-CHOP and had FDG PET/CT for response assessment [136]. iPET negative patients were found to have more favorable PFS, both at interim and EoT periods [136]. The 2-year PFS rates were 87% for EoT PET-negative patients compared to 51% for EoT PET positive patients (*p* 0.001). At interim period, 2-year PFS was 86% for iPET-negative patients compared to 61% for iPET-positive patients, respectively (*p* = 0.0046). Final PET results revealed a significant difference in two-year overall survival as well: 100% versus 88% (*p* = 0.0128) [136]. The results of the previous two studies were explore that vital aspect utilizing FDG PET/CT for response. A retrospective analysis of FOLL05 trial group was carried out by Luminari et al. [137]. The study found that patients who had negative PET scans at EoT had significantly 3-year PFS rates [137]. This suggests that PET scans can be useful in assessing response to treatment in patients with FL [137]. 

To more accurately understand the relationship between FDG PET/CT and survival analysis, Trotman et al. have carried out recent multicentric study [138]. The study was a product of a joint analysis from three prospective studies (PRIMA, PET-FOLLICULAIRE, and FOLL05) [138]. All patient presented with a high tumor burden and were treated with first-line immunochemotherapy [138]. The study found that the EoT PET predicted both PFS and OS [138]. A negative EoT PET was associated with a significantly higher PFS and OS at four years than a positive one [138]. This suggests that the FDG PET/CT at the EoT predicts survival, so a negative study may be a good prognostic indicator for FL patients with high tumor burden. In 2018, a study assessed the prognostic value of EoT PET on a much larger scale, using data from the prospective GALLIUM study [139]. The study compared FDG PET/CT with contrast enhanced CT (CeCT) to determine which one is better for assessing therapy response [139]. Out of all 1202 patients who were enrolled in the study previously, only 595 patients had performed both modalities [139]. All patients were given immunochemotherapy as their first line of treatment and were assessed after finishing therapy [139]. It was found that PET was superior to contrast-enhanced CT for response assessment in FL patients at EoT [139]. More recently, FOLL12 prospective, randomized, open-label multicenter phase III trial was conducted [140]. The aim of this study was to compare a 2-year Rituximab maintenance therapy against a response-adapted therapy approach in FL patients [140]. Response adapted therapy protocol was found to be associated with lower PFS at 2-year interval. It is clear from previous evidence that EoT PET scans can provide accurate predictions of both PFS and OS [140]. 

#### 3.6.2. Chimeric Antigen Receptor Therapy (CAR-T)

The recent approval of axicabtagene ciloleucel for r/r FL was granted after observed results from ZUMA-5 study, which demonstrated an 80% CRR and a 12-month durable response rate of 72% [141]. This offers an effective treatment option for patients who develop refractory disease [142]. A few studies have examined the role of FDG PET/CT in CAR-T for FL patients [136,137,138,139]. These were already mentioned in DLBCL section (CAR-T subheading) as previous studies have pooled aggressive NHL patients together regardless of subtype.

#### 3.6.3. Bispecific Antibodies

More recently, the drug Mosunetuzumab has been approved for the treatment of r/r FL. A recent multicentric phase 2 study has confirmed the efficacy and safety profile of Mosunetuzumab [142]. This is the first in-class approval of a bispecific antibody targeting CD20 and CD3. The activity in FL patients is excellent, with an ORR of approximately 80% and a CR of approximately 60% [142]. However, more studies and research are needed to determine the predictive and prognostic role of FDG PET/CT. Additionally, trials are still ongoing to examine other drugs of the same class.

## 4. Immunotherapy-Related Adverse Effects

Immunotherapy for lymphoma has had great success, but unfortunately, this comes with the consequences of increased Immunotherapy related toxicities. Most of these toxicities are not life-threatening and can be managed if properly diagnosed. Immune-related toxicities observed with FDG PET/CT imaging (Figure 8) will be discussed below.

### 4.1. Monoclonal Antibodies

It is interesting to note that the majority of these side effects are seen in the early stages of therapy. Most frequently, this pattern denotes immune system stimulation. Therefore, there is a high likelihood that these side effects might indicate a positive response to therapy.

#### 4.1.1. Immunotherapy-Related Inflammatory Reactions

Immunotherapy related inflammation can affect any part of the body, as was seen in the ORIENT-1 trial. The purpose of this trial was to investigate the side effects of the anti-PD1 ICI therapy called Sintilimab. Immunotherapy-related adverse events were primarily inflammatory in nature. The most common type of inflammatory event was enterocolitis-related gastrointestinal inflammation, followed by pulmonary involvement and hepatitis [143]. However, there have been no reports of ICI-related fatalities or significant morbidities in this trial, making it unjustifiable to discontinue therapy [143]. Another study concerning ICI adverse events, conducted by Bajwa et al., found that the most commonly reported adverse effects were colitis, hepatitis, and myocarditis [144]. In a study conducted by Petersen et al., FDG PET/CT scans were used to evaluate outcomes and detect related toxicities in patients receiving R-CHOP therapy [145]. It was found that diffuse 18F-FDG uptake in the thyroid can indicate autoimmune thyroiditis and is linked to favorable outcomes [145]. Additionally, another study reported that patients who experience imaging signs of at least one Immunotherapy related adverse event (most commonly colitis or arthritis) through PET/CT are more likely to have more favorable PFS than those who do not have any immunotherapy related adverse events [146]. Therefore, it is advisable to document in the PET/CT report any and all side effects encountered during immunotherapy, even if they are not clinically relevant [147]. In some cases, the results of a FDG PET/CT scan can be misinterpreted, leading to over-diagnosis or uncertainty about the disease process. To avoid this, it is advantageous to concentrate on follow-up and clinical context.

#### 4.1.2. Reactive Changes

One potential indicator of immune system activity in the 18F-FDG PET/CT method of metabolic imaging is the inversion of the liver-to-spleen ratio (normally > 1) [148]. This may suggest immune activation prior to T cell proliferation, as well as reactive nodes in the primary tumor’s drainage basin [149]. This could potentially lead to misinterpretation during assessment. Diffuse reactive splenomegaly and bone marrow activity can occur following treatment with immunotherapy as well as chemotherapy [150]. Usually, these findings do not result in serious consequences but it must be reported and monitored to ensure resolution after therapy discontinuation [151]. Sarcoid-like reactive patterns have also been reported following ICI [152]. FDG-avid bilateral symmetrical hilar and mediastinal lymphadenopathy can occur with or without clinical manifestation [151]. Such patients are usually kept on follow-up to exclude metastatic processes.

#### 4.1.3. Tumor Flare Reaction (TFR)

In the last decade, TFR has been observed as a part of immunotherapy-related adverse effects in lymphoma [153,154,155]. TFR is a clinical syndrome characterized primarily by diffuse FDG-avid generalized lymphadenopathy and splenomegaly, along with other clinical manifestations. It bears a striking resemblance to hyperprogression. The only observed difference between hyperprogression and TFR in lymphoma is the underlying cause. TFR is an immune-mediated process, not a disease progression. This difference can be suspected through FDG PET/CT and confirmed by biopsy. In fact, TFR is more of a clinical manifestation of the pseudoprogression pattern that is sometimes seen during the initial therapy response. Therefore, its incidence should not be used as a reason to discontinue treatment. A case of false-positive FDG PET/CT was reported by Skoura et al. after rituximab therapy [156]. The patient was finally diagnosed with TFR [156]. A PET/CT examination of the patient showed increased metabolic activity of enlarged lymph nodes after R-CHOP treatment and allogeneic transplantation [156]. However, a biopsy of the lymph node revealed extensive reactive T cell infiltration, with no signs of lymphoma cells [156]. Re-examination of PET/CT scans showed no obvious enlargement or increased metabolic activity of lymph nodes after 3 months [156]. More recently, another case report was published describing the clinical and pathological manifestations of TFR [157]. The patient’s TFR was evident after 4 cycles of ICI NHL treatment, as FDG PET/CT showed enlarged and progressing lymphadenopathy above the diaphragm [157]. This was accompanied by other clinical manifestations such as fever, skin rash, joint pain, and poor appetite [157]. After further investigation, TFR was suspected [157]. A left axillary lymph node biopsy was done, ruling out lymphomatous involvement [157]. The patient’s clinical condition improved after the glucocorticoid intake therapy was continued, and the follow-up FDG PET/CT was negative thereafter [157].

### 4.2. Chimeric Antigen Receptor Therapy (CAR-T)

CAR-T cell therapy has been found to cause more immediate and severe side effects than monoclonal antibodies. In order to ensure the best possible outcomes, it is essential to investigate, study, and document these side effects.

#### 4.2.1. Cytokine Release Syndrome (CRS)

CRS is the most common side effect of CAR-T therapy. This syndrome may start on the first day after the therapy transfusion and last up to 9 days. Although FDG PET/CT does not have a direct role in diagnosing CRS, research has indicated that metrics derived from PET can predict the occurrence of CRS. Studies investigating CAR-T for R/R NHL have shown that the tumor burden is strongly correlated with the severity of CRS [158,159,160]. Additionally, recent research has shown that a high tumor burden, as reflected by SUV average (SUVAvg), MTV, and TLG, is a significant risk factor for developing any grade of CRS [127,161,162].

#### 4.2.2. Immune Effector Cell Associated Neurotoxicity Syndrome (ICANS)

The clinical manifestations of ICANS include encephalopathy, delirium, and altered mental status [163,164]. Unlike CRS, the role of FDG PET/CT in ICANS is more widely acknowledged. Brain FDG PET/CT can help exclude differential diagnoses of these cases. In previous studies, neurological deficits associated with ICANS have been observed using brain FDG PET/CT. These deficits typically manifest as areas of reduced metabolism on Brain FDG PET/CT scans. This was demonstrated by Rubin et al., who found that these deficits were associated with CAR-T cell toxicity up to two months after transfusion [165]. In a cohort of six patients, five showed EEG abnormalities that corresponded to hypometabolic areas on PET/CT scans [165]. More recently, Paccagnella et al. described a case of ICANS in a male patient who developed symptoms four days after CAR-T infusion [166]. FDG PET/CT of the brain showed global hypometabolism, particularly in the left hemisphere and frontal regions [166]. Clinical manifestations such as ideo-motor slowing, nominal aphasia, and myoclonic tremor were associated with this condition. However, despite the diffuse slowing seen on EEG, the MRI showed no pathological involvement [166]. After intravenous steroid administration, the patient in the study responded completely [166]. The findings of the current study are consistent with those of a recent case report by Vernier et al. on a patient with DLCBL [167]. PET/CT brain scans performed before and after treatment showed bilateral and diffuse hypometabolism in the parietal and temporal cortex, with no abnormal findings on MRI [167]. These findings suggest that PET/CT is a more sensitive tool than MRI for detecting CAR-T cell-related neurotoxicity.

## 5. Current Challenges and Future Prospects

Many challenges in the field of lymphoma immunotherapy have been overcome through the creation of optimal future plans. This led to the creation of new therapy-specific PET criteria, as well as the exploration of PET-derived metrics for therapy response and outcome prediction.

### 5.1. Current Challenges

#### 5.1.1. Limitations in Low-Income and Conflict Regions

The lack of access to optimal cancer diagnostic services is a major problem in many parts of the world, especially in conflict-affected areas and low-income countries [168]. This can lead to sub-optimal care and a lack of uniformity in cancer diagnosis and treatment [168,169]. Establishing international recovery programs is necessary to overcome these difficulties.

#### 5.1.2. Financial Burden 

Since the FDA approval of ICI and CAR-T therapies, many barriers and considerations have arisen in regards to their widespread use [170]. While some countries are progressing with these therapies, their implementation remains uneven among different nations. To date, many countries are incapable of providing this novel therapies due to lack of financial support, unavailability or financial instability [171]. This is especially apparent in low-income countries and conflict regions [168]. Under these circumstances, medical specialists must carefully select cost-effective treatments for patients and engage them in decision-making to ensure they understand the potential financial consequences of their options.

#### 5.1.3. Volumetric PET Parameters: Promising Yet Overlooked

To date, most institutions rely on semi-quantitative PET parameters through the use of SUVmax values [172]. This is because to SUVmax familiarity among interpreter physicians, validity in medical literature, and appropriate operator reproducibility. However, SUVmax is sensitive to image noise and motion, and its value is dependent on image quality [173]. It should be noted that PET-CT scanners with high spatial resolution tend to produce images with high SUVmax values [173]. Therefore, direct comparison between images produced by different scanners may not be possible [173]. Another factor to consider when interpreting SUV values is that they can be affected by blood glucose levels and uptake time (i.e., the time interval between injection and scanning) [174,175]. On the other hand, volumetric parameters such as MTV and TLG have several advantages over SUV, including being less noise-sensitive. Earlier studies have found that using a variety of PET parameters, including both volumetric and semiquantitative parameters, is an efficient strategy. MTV and TLG have been found to be connected with a higher risk of several different types of cancer [176,177,178,179,180]. Additionally, previous studies found evidence that volumetric indices can be used to predict prognosis [92,104,116,117,121,123,127,161,162]. However, some institutions lack access to these parameters due to limitations in software and experiences [181]. Another issue is the lack of consensus in determining the tumor boundary, which is necessary for reproducibility [173]. Nonetheless, implementing these parameters in future practices can provide additional usefulness once the current barriers are overcome.

#### 5.1.4. Lack of Harmonization in Implementing Novel Response Criteria

Tumor response criteria have been modified over the years to accommodate the recent advancements in lymphoma therapy. Different patterns of response to immunotherapy in lymphoma patients as seen on PET/CT scans necessitated the development of new, therapy-specific criteria [62,65]. However, not all institutions have adopted these new criteria. Many instead are still relying on the Lugano criteria which does not take into account pseudoprogression [182]. There is therefore a need to focus on harmonizing response assessment in order to optimize response evaluation.

### 5.2. Future Prospects 

#### 5.2.1. FDG Alternatives

The current focus in lymphoma treatment is on providing targeted therapy using theranostics. The CXC chemokine receptor 4 (CXCR4) is a promising cancer treatment target. This transmembrane receptor is involved in hematopoietic stem cell migration and homing to the bone marrow [183]. Multiple studies have demonstrated the imaging and targeting capabilities of 68Ga-Pentixafor for CXCR4-expressing lymphomas [184,185]. Fibroblast activation protein inhibitor (FAPI) is currently being studied for its potential use as both a diagnostic and therapeutic tool in lymphoma. Preliminary results are encouraging in terms of safety and efficacy [186]. FAP is overexpressed by cancer-associated fibroblasts present in the tumor microenvironment, which leads to high tumor uptake and very low accumulation in normal tissues, achieving excellent results [187]. Incorporation of ^18^Fluorine and Fludarabine (a chemotherapy drug already used in low-grade lymphomas) was examined in variable and low FDG-avid lymphomas. Studies in animals and humans have demonstrated lower uptake of these drugs in inflammatory cells compared to FDG and a better correlation with histology than the latter [188]. The sensitivity of ^18^F-Fludarabine for detecting indolent lymphoma lesions has been found to be good, with reports indicating that it could potentially be used to replace FDG PET/CT imaging for such cases [188,189,190,191].

#### 5.2.2. Radiomics and Machine Learning

The use of FDG PET/CT for predictive and prognostic purposes in lymphoma has been well established over the past few decades [26,68]. However, many other features of this imaging modality have only recently been explored [26]. Machine learning algorithms can be used to detect and measure overall tumor burden and are currently being examined. Frood et al. investigate the potential benefits of using a radiomic model to indicate 2-year PFS of HL patients [192]. The study found that the model was useful, but that more research is needed to confirm efficacy [192]. Radiomics can be used to minimize time and effort during PET/CT assessment. Radiomic-based segmentation techniques studied by Sollini et al. can serve as a useful tool in clinical practice for assessing involved lesions in HL patients, with an overall accuracy of 82% [193]. This radiomic technique is based on the utilization of lesion similarity analysis when assessing involved lesions in HL patients. These methods have shown to be useful in histological prediction, prognostic evaluation, and the definition of bone marrow involvement [193]. Nonetheless, more clinical research is necessary to determine the efficacy of these AI algorithms [26].

## 6. Conclusions

Cancer immunotherapy continues to deliver alternative and superior therapy choices for certain stages and subtypes of lymphoma. Advancement in this field necessitates the development of optimal response criteria that can provide predictive and prognostic information through the use of metabolic parameters. FDG PET/CT plays a key role in the newly developed response criteria, and several FDG PET/CT-based criteria have been proposed to address all patterns of response to therapy, including indeterminate response, pseudoprogression, and hyperprogression using several metrics, such as SUV, MTV, and TLG. Immunotherapy-related side effects should be taken into account and not misinterpreted as disease progression; moreover, they may predict treatment effectiveness. Certainly, more harmonization and consensus are still desired, and the future holds promise for superior immunotherapies and more optimized criteria for assessing response and impacting patient outcomes.

## Figures and Tables

**Figure 1 cancers-15-01063-f001:**
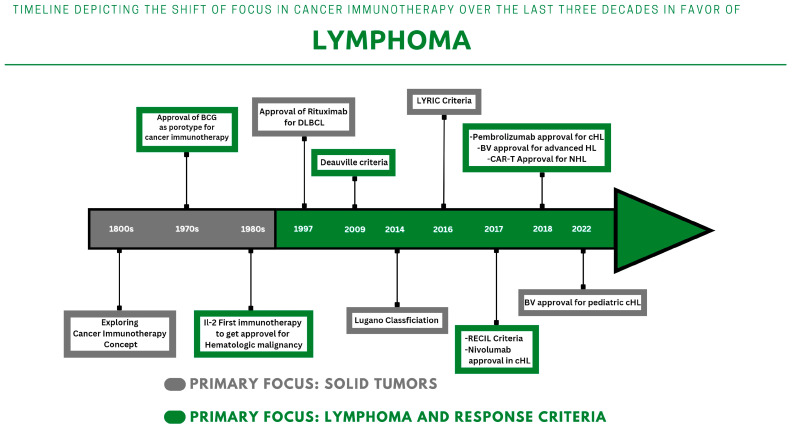
Timeline Depicting the shift of focus in cancer immunotherapy over the last three decades in favor of lymphoma.

**Figure 2 cancers-15-01063-f002:**
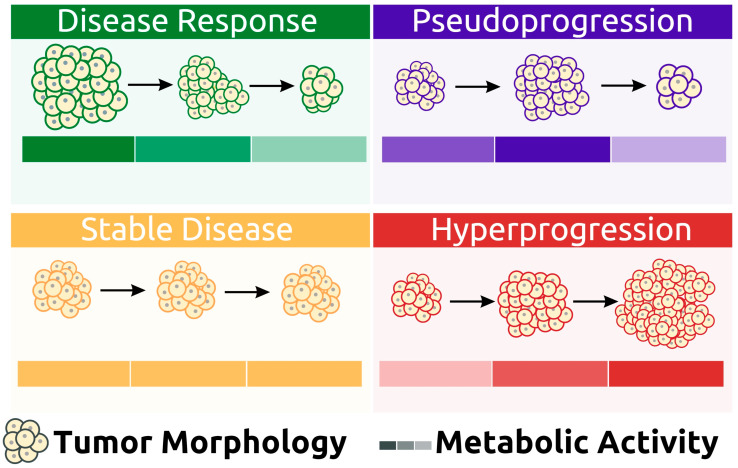
Graphical presentation of different response patterns achieved after lymphoma immunotherapy administration.

**Figure 3 cancers-15-01063-f003:**
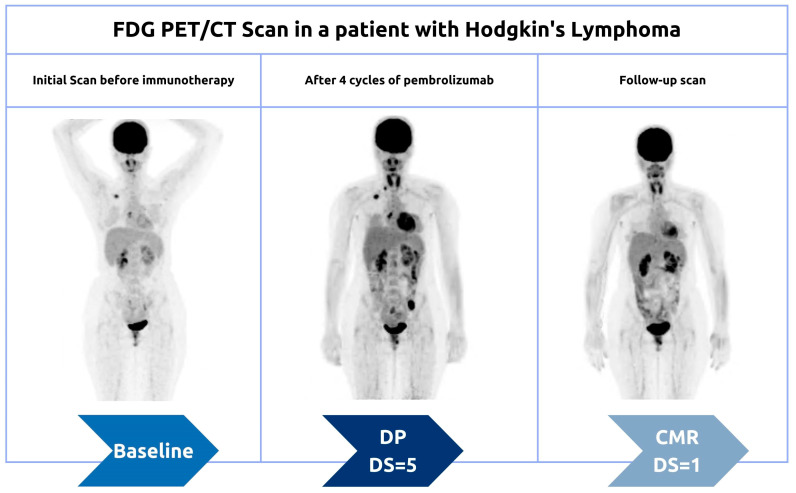
Serial Maximum Intensity Projection (MIP) images of an HL patient demonstrating pseudoprogression pattern during the interim period.

**Figure 4 cancers-15-01063-f004:**
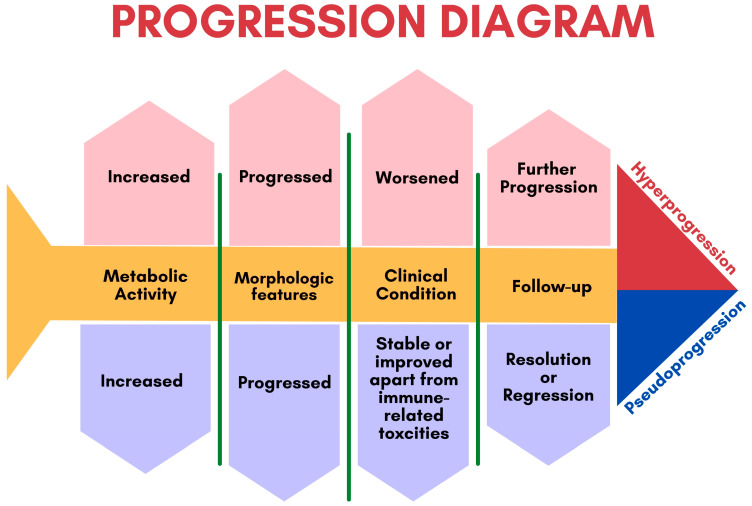
A fish bone diagram demonstrating the difference between pseudoprogression and hyperprogression.

**Figure 5 cancers-15-01063-f005:**
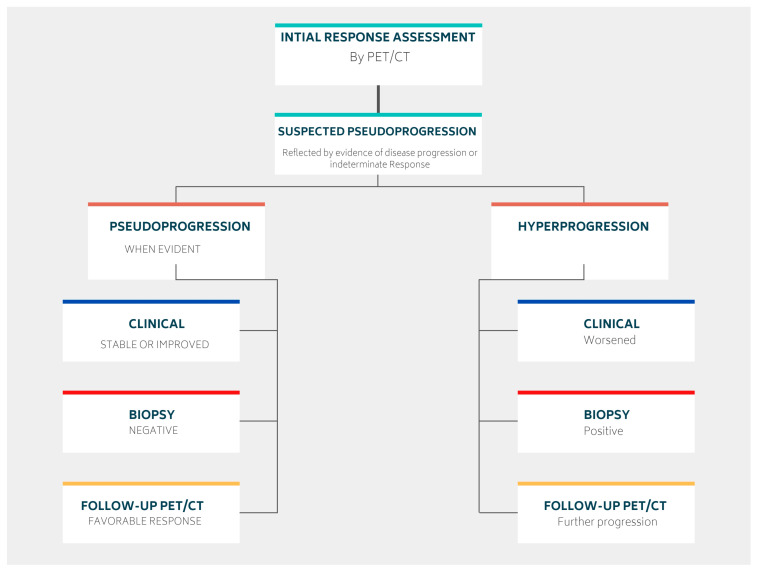
Schematic showing clinical, histopathological, and imaging differences between pseudo-progression and hyperprogression.

**Figure 6 cancers-15-01063-f006:**
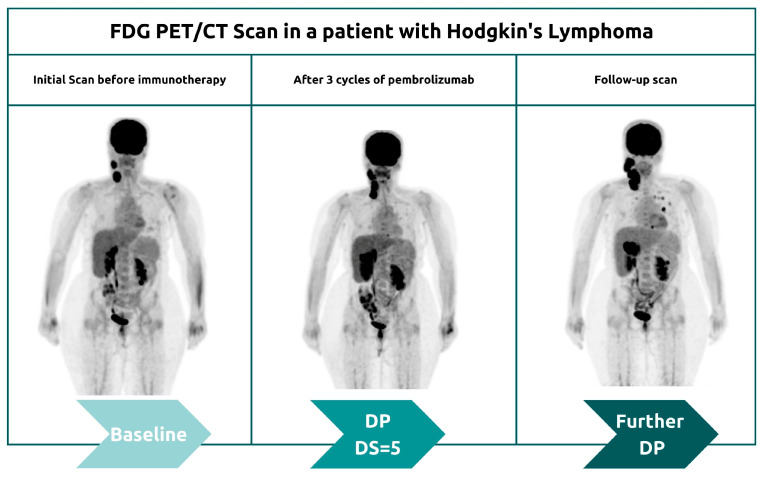
Serial Maximum Intensity Projection (MIP) images of an HL patient demonstrating a hyperprogression pattern.

**Figure 7 cancers-15-01063-f007:**
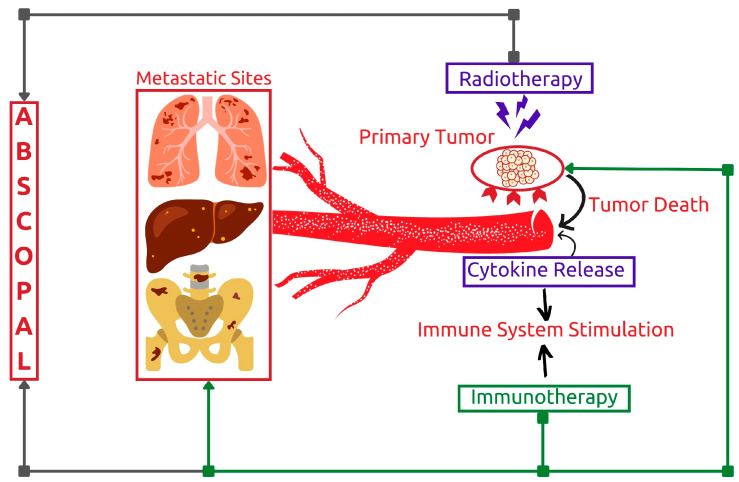
Graphical presentation of proposed abscopal effect mechanism.

**Figure 8 cancers-15-01063-f008:**
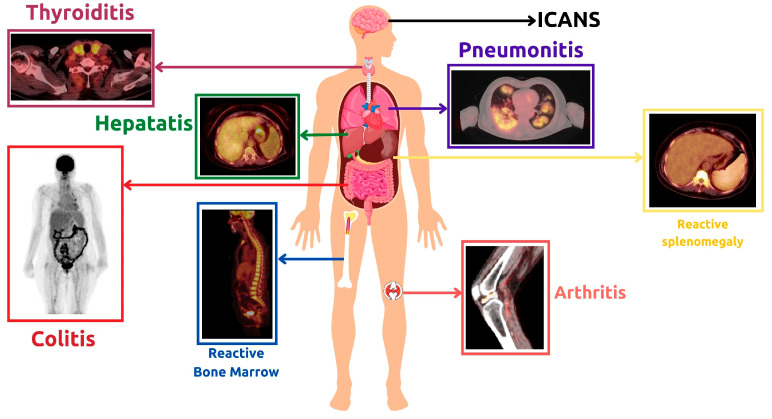
Clinical examples of Immunotherapy-Related Adverse Effects observed during FDG PET/CT imaging in lymphoma patients receiving immunotherapy.

**Table 1 cancers-15-01063-t001:** List of approved immunotherapies for lymphoma.

Drug Name	Class	Main Action	Treatment Protocol	Approved for
Rituximab	mAb ^1^	CD-20 Antibody	With chemotherapy	First line for NHL ^2^
Brentuximab Vedotin	mAb ^1^	CD-30 Antibody	With chemotherapy	Advanced HL ^3^
Nivolumab	ICI ^4^	PD-1 Blockade	Standalone	cHL ^5^
Pembrolizumab	ICI ^4^	PD-1 Blockade	Standalone	Refractory cHL ^5^
Tisagenlecleuce	CAR-T ^6^	T-lymphocyte-mediated CD-19 expression	Standalone	Adult R/R DLBCL ^7^
Lisocabtagenel maraleuecel	CAR-T ^6^	T-lymphocyte-mediated CD-19 expression	Standalone	R/R large B-cell lymphoma
Mosunetuzumab	BiTes ^7^	Follicular Lymphoma	Standalone	R/R Follicular Lymphoma

^1^ mAb: Monoclonal Antibody; ^2^ NHL: Non-Hodgkin’s Lymphoma; ^3^ HL: Hodgkin’s Lymphoma, ^4^ ICI: Immune Checkpoint Inhibitor; ^5^ cHL: Classical Hodgkin’s Lymphoma; ^6^ CAR-T: Chimeric Antigen Receptor Therapy; ^7^ BiTes: bispecific T-cell engagers.

**Table 2 cancers-15-01063-t002:** Lugano classification and deauville 5-point scale (D5PS) in FDG-avid lymphomas.

**Deauville 5-Point Scale (5PS)**	
DS *1	No uptake
DS2	Uptake ≤ mediastinum
DS3	Uptake > mediastinum but ≤ liver
DS4	Uptake moderately higher than liver
DS5	Uptake markedly higher than liver and/or new lesions ^+^

* DS: Deauville Score; ^+^ Scores 4 and 5 are defined as uptake > the maximum standardized uptake value (SUV) of the liver and ≥2–3 x the maximum SUV of the liver, respectively.

**Table 3 cancers-15-01063-t003:** Status and degree of FDG avidity in each type and subtype of Lymphoma.

Category	Subtype of Lymphoma	FDG Avidity	Degree of FDG Avidity
HL ^1^	Classical	Avid	High
	Mixed cellularity	Avid	Moderate to high
	Lymphocyte depletion	Avid	Moderate to high
	Lymphocyte predominance	Avid	Moderate
Aggressive NHL ^2^	Diffuse large B-cell	Avid	High
	Burkitt	Avid	High
	Anaplastic Large cell	Avid	High
	Mantle Cell	Avid	Moderate
Indolent NHL ^2^	Follicular	Variable	Low-high
	Lymphoplasmacytic	Variable	Low-high
	Marginal zone	Variable	None-high
	Small lymphocytic	Variable	None-high
	Cutaneous Anaplastic	Variable	None-moderate

^1^ HL: Hodgkin’s Lymphoma; ^2^ NHL: Non-Hodgkin’s Lymphoma.

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
