# Peer review of "FDG-PET/CT in the Monitoring of Lymphoma Immunotherapy Response: Current Status and Future Prospects"

_cancers, 2023, doi:10.3390/cancers15041063_

Round 1
Reviewer 1 Report
The authors reported a review on the role of FDG-PET/CT in the monitoring of lymphoma immunotherapy response. The review is interesting but there are some pitfalls that deserve some comments to improve the manuscript.
The main focus of the manuscript should be the role of PET in the evaluation of the response i.e the pros and contras, false positive results, pitfalls etc. It means that the first part of the manuscript is out of the scope because it describes some clinical findings in a largely incomplete summary not reporting a lot of data, missing some compounds as axi-cell etc . I think it would be better reducing this part to a short introduction.
Pseudoprogression is one of the most intriguing issues with FDG-PET/CT during immunotherapy in lymphomas. This part should be better explained, and it would be also helpful for the clinicians to add some hints how to manage these situations.
Hodgkin’s Lymphoma. Interim PET is the most important predictive factor in HL patients treated with chemotherapy (mainly ABVD i.e PET after two courses). The authors should report and discuss how the value of interim PET has changed incorporating immunotherapy (Brentuximab-vedotin and/or ICI) with chemotherapy or with immunotherapy alone.
Non-Hodgkin’s lymphoma. This paragraph should be completely rewritten. First the authors should report separately the data for follicular lymphoma and DLBCL. Either the role of PET as interim evaluation and end of treatment should be discussed for both entities with pros and contras. The authors should cite the most recent papers in this field, i.e the Gallium data or the FOL12 data for FL and the Goya data for DLBCL that are lacking in the manuscript. Moreover, the authors should discuss also the value of semiquantitative PET parameters in DLBCL. Indeed, the authors state that interim PET in DLBCL is a valid tool but there are also papers that did not find a prognostic value of interim PET. Please discuss this point in a more critical and balanced way. The sentence at line 380 page 11 is overemphasized because response-adapted management approaches both in FL and DLBCL are not validated yet and there are some studies that failed to show a benefit with such approach.
Bispecific antibodies as Mosunetuzumab, Glofitamab and epcoritamab are an emerging, important immunotherapy in NHL and Mosunetuzumab is already FDA/EMA approved and Glofitamab and Epcoritaman are on the way. Data on the role of PET/CT and the possible pitfalls with these treatments should be added.
CART in DLBCL. Line 423 page 12. Which robust data support the statement that is”essential to have adequate control prior to therapy initiation…”? It is true that a better control of the disease prior CART reduces the risk of toxicity but the data on the response are contradictory because a prolonged bridging therapy may reduce the efficacy of CART treatment favoring the emergence of a more resistant disease. Please rephrase this statement incorporating these items with a more balanced conclusion.
The list of references is too long, many of them are too old and can be deleted.
Author Response
The authors reported a review on the role of FDG-PET/CT in the monitoring of lymphoma immunotherapy response. The review is interesting but there are some pitfalls that deserve some comments to improve the manuscript.
Thank you very much for this informative review shared along with respectful review points and vital questions that necessitate further improvement in many aspects of our review article to cover each and every aspect of this vital subject.
Kindly note that we have chosen to subdivide each question with number labeling to facilitate the review process and answer each question accordingly .
Below are the answers to your respectful review points.
- The main focus of the manuscript should be the role of PET in the evaluation of the response i.e the pros and contras, false positive results, pitfalls etc. It means that the first part of the manuscript is out of the scope because it describes some clinical findings in a largely incomplete summary not reporting a lot of data, missing some compounds as Axi-cell etc. I think it would be better reducing this part to a short introduction.
We appreciate you bringing this up.. The first part of the manuscript, which at first glance, seems not to be in line with the primary aim of this review. Actually, this part was challenging for us to include in this review, but it was required for a number of reasons. It was hard to go into detail for each class, but it is still important to highlight some points so that readers (particularly those coming from nuclear medicine and radiology background) can get a general idea of how immunotherapies work, why they are approved by the FDA, and what the future holds for promising immunotherapies for lymphoma.
In response to your respectful input, we have made some considerable changes in this segment to make it more appealing and to further correct some of the previously introduced incoherence. These include:
- Trimming down previous segments on cancer immunotherapy that are irrelevant to our subject (current word count is 793 vs 939 previously).
- Re-ordering immunotherapy classes (irrelevant classes were trimmed down)
- Changing the title from "Immunotherapy: classification and development" to "Immunotherapy in lymphoma"
- Removing outdated references
- Directing the focus toward ICI and CART (subjects of interest).
- Adding new segment labeled as “Bispecific antibodies”. Kindly check lines from 135-140.
- B. From a nuclear medicine and medical imaging perspective, we think the current information is still needed to get an orientation about immunotherapy classes as a comprehensible background about the subject. Axi-cell is mentioned briefly in the last paragraphs to denote its importance as a part of the CAR-T.
- Pseudoprogression is one of the most intriguing issues with FDG-PET/CT during immunotherapy in lymphomas. This part should be better explained, and it would be also helpful for the clinicians to add some hints how to manage these situations.
The pseudoprogression segment has undergone a number of refinements and updates in response to your respectful review. In addition, more figures and tables have been added to provide a clinical perspective on management and pathophysiology. These include:
- The whole segment was completely refined and rewritten to express more data in regards to imaging and clinical perspective along with up-to-date reference citation to strengthen the point of view.
- A new diagrams were added figure 4 and 5} to illustrate the current trend in management of such cases.
- Kindly check new changes applied from line 163-184.
- B. This subject was also mentioned briefly as a part of immune related side effects, labeled as “Tumor Flare Reaction” in page lines 634-658.
- Hodgkin’s Lymphoma. Interim PET is the most important predictive factor in HL patients treated with chemotherapy (mainly ABVD i.e PET after two courses). The authors should report and discuss how the value of interim PET has changed incorporating immunotherapy (Brentuximab-vedotin and/or ICI) with chemotherapy or with immunotherapy alone.
Thank you for bringing this important point to our attention. For such novel agents. adding this information to our discussion about Hodgkin lymphoma will help strengthen our argument and provide important knowledge to the reader. We have included recent evidence found in the previous literature about the role of PET/CT in such scenarios at interim periods. These include:
- Adding short segments about the value of interim PET in revolutionizing treatment protocol in HL patients incorporating ICI as part of the therapy provided. Kindly check new changes from line 315-323.
- Adding short segments about the value of interim PET in revolutionizing treatment protocol in HL patients incorporating BV as part of the therapy provided. Kindly check new changes from line 338-365.
- Non-Hodgkin’s lymphoma. This paragraph should be completely rewritten.
On this point, we followed the reviewer's guidance. As a result, the Non-Hodgkin Lymphoma section was divided into three distinct sections to demonstrate each subtype separately. Thus, the non-Hodgkin lymphoma part was subdivided into 3 separate segments to demonstrate each subtype separately. This was done following your respectful comment about this important part of our review article. We have tried our best to answer and involved each and every review points you have mentioned in the following points. Before that we would like to thank you for making this effort to optimize our manuscript in the best way possible.
- First the authors should report separately the data for follicular lymphoma and DLBCL.
We have already discussed each and every subtype of Non-Hodgkin Lymphoma as a separate segment.
- Either the role of PET as interim evaluation and end of treatment should be discussed for both entities with pros and contras.
The role of PET/CT for both interim and end of therapy intervals have been discussed in the recent updated version. You can find evidence to this update in DLBCL and Follicular lymphoma Sections highlighted in grey.
- The authors should cite the most recent papers in this field, i.e the Gallium data or the FOLL12 data for FL and the Goya data for DLBCL that are lacking in the manuscript.
It was very kind of you to provide this important and up-to-date references to include in our review article. In response, we have included these important references in our updated version. You can find the new paragraphs In lines (442-450 for GOYA, 556-564 for GALLIUM, and 564-569 for FOLL12).
- Moreover, the authors should discuss also the value of semiquantitative PET parameters in DLBCL.
We have provided the value of semiquantitative PET parameters in DLBCL section including the data analyzed the from GOYA group patients. You can find this added paragraphs in updated DLBCL section.
- Indeed, the authors state that interim PET in DLBCL is a valid tool but there are also papers that did not find a prognostic value of interim PET. Please discuss this point in a more critical and balanced way.
Kindly review the recent changes made in rituximab segment within DLBCL heading to track the changes made in response to your respectful review point. This can be found in lines 438-450.
- The sentence at line 380 page 11 is overemphasized because response-adapted management approaches both in FL and DLBCL are not validated yet and there are some studies that failed to show a benefit with such approach.
Apologies for the previous misinterpretation. The new paragraph was updated to present the idea in a moderate manner. Kinly track the new changes in line 456-457.
- Bispecific antibodies as Mosunetuzumab, Glofitamab and Epcoritamab are an emerging, important immunotherapy in NHL and Mosunetuzumab is already FDA/EMA approved and Glofitamab and Epcoritaman are on the way. Data on the role of PET/CT and the possible pitfalls with these treatments should be added.
We have added up to date knowledge about this class of cancer immunotherapy. This can be found in lines 136-141 and also in lines 580-587. To date, little data exists on the role of bispecific antibodies, there is still relatively little information available on this topic, particularly in regards to PET/CT.
- CART in DLBCL. Line 423 page 12. Which robust data support the statement that is”essential to have adequate control prior to therapy initiation…”? It is true that a better control of the disease prior CART reduces the risk of toxicity but the data on the response are contradictory because a prolonged bridging therapy may reduce the efficacy of CART treatment favoring the emergence of a more resistant disease. Please rephrase this statement incorporating these items with a more balanced conclusion.
The previous misinterpretation was corrected in the updated version replacing the previous conclusion to a more balanced statement. Kindly track changes in lines 503-504.
- The list of references is too long, many of them are too old and can be deleted.
Many outdated references were deleted in the new version using only up to date evidence from literature.

Reviewer 2 Report
The present paper is aimed at exploring cancer immunotherapy with a focus on lymphoma. In particular, the role of FDG PET/CT in immunotherapy of FDG avid lymphoma has been discussed, also addressing patterns of immunotherapy-related toxicities and future directions in the field. The paper is well written, it is complete and informative for the readers, reporting a comprehensive classification of immunotherapy and a description of the role of FDG PET in lymphoma in NHL and HL. I would suggest to add some PET figures of different clinical settings (LH, LNH, hyperprogression, pseudoprogression…etc…) to show clearly the different imaging patterns.Author Response
Thank you very much for this informative review shared along with instructive review points that necessitate further improvement in some aspects of our review article to cover each and every aspect of this vital subject. In response to your respectful review, we have added a new figure [Figure 6] to demonstrate hyperprogression with additional 2 diagrams [Figure 3 and 4] to provide more knowledge and to strengthen the point of view shared in the text template.

Reviewer 3 Report
The Authors have performed a review on the role of immunotherapy in patients with lymphoma. The manuscript is very well written and comprehensive of all aspects on this complex topic.
Only a minor change. It should be preferable to add this sentence in the conclusions section: "immunotherapy-related side effects should be taken into account and not misinterpreted as disease progression; moreover, they may predict treatment effectiveness"
Author Response
Thank you for your comments. It was very inspiring to recommend our work on this review and to provide instructive comments that were used to reinforce and strengthen the section on conclusions. The conclusions section is one of the most important aspects of the paper, and your input has helped to make it even more impactful. In response to your respectful review, we have added the requested paragraph in conclusion section. Kindly track changes found in lines 788-790.

Reviewer 4 Report
Cancer immunotherapy continues to deliver alternative and superior therapy choices for certain stages and subtypes of lymphoma.
Advancement in this field necessitates the development of optimal response criteria that can provide predictive and prognostic information through the use of metabolic parameters.
FDG PET/CT plays a key role in the newly developed response criteria and several FDG PET/CT-based criteria have been proposed to address all patterns of response to therapy.
Including several response types such as indeterminate response, pseudoprogression and hyperprogression using several metrics, such as SUV, MTV and TLG.
Certainly, more harmonization and consensus are still desired, and the future holds promise for superior immunotherapies and more optimized criteria for assessing response and impacting patient outcomes.
Review. Cancers. 2126045
Q 1. Title
Is short, very complete and fully informative
Q 2. Abstract and Keywords
Cancer immunotherapy has been extensively investigated in lymphoma over the last three decades. This new treatment modality is now established as a way to manage and maintain several stages and subtypes of lymphoma. The establishment of this novel therapy has necessitated the development of new imaging response criteria to evaluate and follow up cancer patients. Several FDG PET/CT-based response criteria have emerged to address and encompass the various most commonly observed response patterns. Many of the proposed response criteria are currently being used to evaluate and predict response. The purpose of this review is to address the efficacy and side effects of cancer immunotherapy, and to correlate this with the proposed criteria and relevant patterns of FDG PET/CT in lymphoma immunotherapy as applicable. The latest updates and future prospects in lymphoma immunotherapy, as well as PET/CT potentials, will be discussed.
Keywords have been well selected and are very complete for searching
Q 3. 1. Introduction
Lymphoma immunotherapy is becoming more appealing over time, as evidenced by the wide variety of approved therapy options that are now available for certain stages and subtypes of lymphoma. Applying this therapy in clinical practice has broadened the concept of lymphoma treatment, as it can fight tumor biology in addition to on-site disease eradication. Management of lymphoma has seen great advances in recent years, with a shift in focus to immunotherapy in the last three decades. This has translated to FDA approval of several immunotherapies. Positron emission tomography coupled with computed tomography (PET/CT) allows for better evaluation of response to immunotherapy in FDG-avid lymphomas, as well as providing prognostication insights. Metabolic PET parameters are reliable predictors in the context of absent alternative biomarkers.
This paper will explore cancer immunotherapy with a focus on lymphoma. Specifically, it will discuss the role of FDG PET/CT in immunotherapy of FDG avid lymphoma. Finally, it will address patterns of immunotherapy-related toxicities and future directions.
2. Cancer immunotherapy: Classification, Previous and Current Facts
2.1. Historical overview of cancer immunotherapy
The concept of cancer immunotherapy has been explored and studied since the 19th century. However, its clinical implementation remained debatable until the approval of the first immunotherapy drug in 1976 [Figure 1]. The first generation of immunotherapy relies on the action of vaccines to boost the immune response. This was followed by the utilization of anti-tumor cytokines, monoclonal antibodies, oncolytic viruses and adoptive cell therapies to recruit immune cells against certain types of cancers. Thus far, there are many types of cancer immunotherapies that are implemented in lymphoma treatment.
2.2. Immunotherapy: classification and development
2.2.1. First generation immunotherapy: The old and cornerstone
After years of research and experimentation, it became evident in the last century that certain bacterial vaccines, such as Bacille Calmette-Guérin (BCG), could recruit immune cells to inhibit the recurrence of urinary bladder cancer. This approach remains active and is still adopted in clinical practice. Currently, there are several types of vaccines that play a vital role in preventing cancer.
2.2.2. Cytokines: the following milestone
Interferons (IFN) were discovered in 1957 as part of early milestones in immunotherapy development. After years of research, IFN-α was approved for treatment of hairy cell leukemia in 1986. This family of cytokines were effective in inhibiting tumor cell proliferation and enhancing cancer apoptosis. The timeframe between interferon discovery and adoption witnessed the discovery of interleukins (IL), namely IL-2. IL-2 were found to be effective in treating advanced renal cell carcinoma (RCC) and metastatic melanoma. The Food and Drug Administration (FDA) approval for utilization in advanced RCC and metastatic melanoma were granted in years 1991 and 1998 respectively.
2.2.3. Monoclonal antibodies: from rituximab to immune checkpoint inhibitors
Immunomodulating antibodies were studied extensively in the 1990s and rituximab emerged as a prototype for many other monoclonal antibodies in 1997. It is incorporated with cyclophosphamide, doxorubicin HCl, vincristine, and prednisone in the RCHOP protocol for the treatment of diffuse large B-cell lymphoma (DLBCL). Sometimes it is added to other anthracycline-based chemotherapeutics for NHL treatment
The adoption of the immunochemotherapy protocol (R-CHOP) has been shown to have a much higher survival impact when compared to standard chemotherapy protocols.
Under the same umbrella immune check point inhibitors (ICI) became approved and available for many cancer types. FDA approved the use of ipilimumab in 2011 as a therapy for advanced melanoma. Ipilimumab was the first ICI drug to get FDA approval for melanoma. In 2016, Nivolumab attained the same approval as the first programmed cell death protein 1 (PD-1) inhibitor for the treatment of Hodgkin’s lymphoma (HL). Within the following year, another anti-PD-1 inhibitor (pembrolizumab) got approved. The current range of ICI drugs use checkpoint blockade as their primary mode of action, yet there are distinctions between the various subclasses. The effects of each IC class differ depending on their respective target checkpoints. As a result, only certain types of cancers are responsive to the various ICI subtypes. To date, only a limited number of ICIs have been approved for clinical use by the FDA [Table 1], with others likely to follow in the foreseeable future. The ICI has been shown to be effective in clinical settings against HL cells and the tumor microenvironment . The programmed death ligand-1 programmed cell death protein (PD-L1) is potentially blocked and inhibited through ICI administration, which was observed in 70% of cases. This inhibitory pathway can terminate tumor growth and stimulate the immune system against HL cells.
More recently, FDA approval of Brentuximab Vedotin (BV)was attained in 2018 . Approval was gained after successful results obtained from randomized clinical trial of 1,334 patients that have received either BV plus doxorubicin, vinblastine, and dacarbazine (BV + AVD) or bleomycin plus AVD (ABVD). At first, BV was granted approval to treat advanced adult HL followed by recent FDA approval of pediatric untreated classical HL
2.2.4. Oncolytic virotherapy: Reinforcing passive immunotherapy
As cancer immunotherapy advances, researchers are investigating new ways to harness the power of the immune system to fight cancer. One promising area of research is using viruses to target cancer cells and boost the immune system's response to the tumor environment. This approach has shown success in treating melanoma, with genetically modified herpes viruses approved for use since 2015.
2.2.5. Adoptive cell therapy: the most recent addition to the group
In recent years, researchers have become more interested in targeting both genetic and cellular abnormalities in tumors in order to better control cancer growth and spread. A new type of T cell therapy, known as chimeric antigen receptor therapy (CAR-T), has emerged as a promising treatment option. In this therapy, T cells are taken from a patient's blood and modified to express artificial receptors that are specifically targeted at a particular tumor antigen. This allows the T cells to bind to and kill the cancer cells while leaving healthy cells unharmed. In detail, the patient's T-cell will be equipped with an artificial CAR. These receptors are composed of an antibody-derived single-chain variable fragment, a transmembrane, and a signaling domain. The CAR segment will allow T-cells to target tumor antigens. Through antigen binding, the CAR will induce cytokines recruitment and proliferation against receptor-specific cancer cells.
CAR-T cells have been divided into four generations, depending on the number of intracellular signaling molecules involved. The first generation of CAR-T cells uses only the CD3ζ chain as its single intracellular signaling domain. Subsequent generations of CAR-T cells have incorporated one or two co-stimulatory signaling domains, such as the CD28 or CD137 (4-1BB), to improve antitumor activity. The fourth generation of "armored" CAR-T cells has been developed to incorporate additional co-stimulatory ligands or cytokines in order to improve the efficacy of CAR-T cells. CAR-T cells were approved for use in leukemia in 2017 and then for lymphoma in the following year. CAR-T cells represent a major advancement in the field of immunotherapies. This is evident from the fact that two CAR-T cells, Yescarta (axicabtagene ciloleucel) and Kymriah (tisagenlecleucel), are commercially available
3. Role of PET/CT in lymphoma Immunotherapy
The lack of reliable biomarkers to measure immunotherapy response has made FDG PET/CT response criteria more useful. This hybrid imaging modality can use various metabolic parameters to predict and evaluate therapy response. In fact, FDG PET/CT is the only imaging modality with the ability to evaluate therapy response and demonstrate metabolic aspects of immunotherapy related side effects
3.1. Response Patterns
The response to immunotherapy varies among patients, with most experiencing a good initial response. However, some patients may demonstrate different patterns of response after therapy administration [Figure 2].
3.1.1. Pseudoprogression
Some patients may initially appear to experience disease progression (known as pseudo-progression) before ultimately achieving favorable clinical picture]. This false positive pattern was first reported in 15% of patients receiving anti-CTLA4 therapy. The need to correct initial erroneous positive results necessitate the implementation of new response criteria. Pseudo-progression was first observed in solid tumors and later reported in lymphomas, introducing additional confusion in PET-driven response assessment. Rather than being indicative of actual progression, pseudo-progression is more like a flare phenomenon caused by massive immune stimulation. During the initial phases of therapy, immune cells may be recruited into the tumor micro-environment, leading to a temporary increase in tumor size and metabolic activity. However, pseudo-progression can be confirmed during follow-up imaging through eventual tumor regression and favorable clinical outcome.
3.1.2. Hyperprogression
More recently, a permanent progression has been observed and evidenced by the increased rate of tumor growth. This was supported by Phase 3 clinical trials which demonstrated decreased survival outcome in some patients who underwent immunotherapy treatment. Under the category of hyperprogression, this pattern most commonly affects elderly patients and has been noted in up to 29% of patients receiving immunotherapy. In hyperprogression, there is evidence of dramatic tumor growth rate associated with clinical worsening . When comparing baseline imaging with initial therapy imaging, there is a minimum of twofold increase in overall tumor burden . In such cases, the only choice is to terminate immunotherapy.
3.1.3. Potentiating Abscopal Effect
The findings suggest that combining radiotherapy with immunotherapy may boost the abscopal effect of local radiotherapy treatment . This response pattern was first observed in the 1950s, after researchers noted clinical responses at distant metastatic sites following administration of locoregional radiotherapy . Later research showed that this phenomenon is mediated by T cells, and that the incidence of abscopal effect is favorable in immunocompetent patients. Enhancing immune system response through immunotherapies can therefore result in a potential synergistic effect. Researchers are still working to determine the exact mechanism of this effect, after several reported clinical cases.
3.2. PET Response Criteria in lymphoma
Given the lack of biological markers to assess the efficacy of immunotherapy . It was necessary to create therapy-specific criteria to assess the wide array of response patterns encountered.
3.2.1. Lugano classification
The Lugano criteria are widely used in studies and clinical trials of immunotherapy drugs, despite being non-specific for immunotherapy response. The criteria provide a solid foundation for future therapy-specific response criteria. In 2014, the Lugano classification was adopted by a team of specialists in oncology, hematology, radiology, and nuclear medicine . This classification uses metabolic PET parameters to assess response to therapy at the end of therapy (EoT) and during the interim period (iPET) . Since then, it has been considered the gold standard interpretation criteria for FDG-avid lymphomas . The Lugano study introduced a five-point scale for assessing metabolic response, instead of using a dichotomous response pattern. The ordinal scale, consisting of five Deauville scores [Table 2], was used to examine the degree of response. The degree of response can be measured by qualitative visual assessment of FDG uptake within the most intense residual lymphomatous lesion identified during EoT or i-PET. The retrieved values are then visually compared to reference metabolic values derived from background, mediastinal blood pool, and liver. Complete metabolic response is indicated if the Deauville 5-point scale (5PS) is between 1-3. A score of 4 or 5 on the five-point scale, with evident reduced uptake since the baseline study, is indicative of partial metabolic response (PMR). A DS of 4 or 5 can also indicate disease progression (PD) when there is evidence of increased uptake since the baseline study, with or without the interval development of new FDG avid lymphomatous disease process.
3.2.2. Lymphoma Response to Immunomodulatory Therapy Criteria (LYRIC)
The introduction of immunotherapy has necessitated modifications to the previously established Lugano classification in order to account for pseudo-progression. This transient pattern of response, characterized by an increase in tumor size and metabolic flare, can make it difficult to determine if a patient is truly progressing or if their disease is simply pseudo-progressing. In 2016, the LYRIC group introduced a new category of response, labeled as Indeterminate Response (IR), to account for this pseudo-progression pattern. If there is suspicion of pseudo-progression, a patient can be classified as IR and therapy can be continued for up to 12 weeks before a definitive confirmation is made [Table S1]. After that point, follow-up FDG PET/CT imaging can be used to discriminate between true progression and pseudo-progression. Additionally, histopathologic confirmation can be pursued to better understand the disease pattern.
3.2.3. Response evaluation criteria in Lymphoma (RECIL)
In an effort to harmonize lymphoma response criteria in clinical trials, RECIL revolutionized the way to evaluate immunotherapy. Anatomically, lesions measurement was modified to include only unidimensional measurement of the long diameter of 3 selected target lesions . In addition, RECIL proposed measuring and comparing difference in tumor burden. As a result, complete response (CR) has replaced CMR and would require at least a 30% reduction in tumor burden in addition to a DS range of 1-3. In a parallel fashion, a higher DS of 4-5 would indicate a PR, achieving a reduction in tumor burden of at least 30% . A new category labeled as minor response has been proposed in cases of at least 10% tumor burden reduction, not exceeding the 30% threshold. A stable disease pattern can be observed if the range of change in the tumor burden lies between -10% and +20%. Otherwise, disease progression implies a value of more than 20% increase in tumor burden, with or without the appearance of a new lesion . It is noteworthy that both PD and minor response categories do not require correlation with 5PS . Additionally, disease relapse is considered when a newly appearing lesion exceeds 1.9 cm in the long axis.
3.3. FDG PET/CT in Hodgkin’s Lymphoma (HL)
HL is known for its high FDG avidity and chemosensitivity . Previous research has focused on exploring standard therapeutic protocols to determine therapy response and outcome. However, with the emergence of new lines of cancer immunotherapy, new treatment protocols have been introduced, accompanied by new PET/CT response criteria specific to lymphoma immunotherapy.
3.3.1. Immune Checkpoint Inhibitors
The idea behind the ICI mechanism has led to the first clinical trial conducted by Ansell et al. . This first clinical trial reported a response rate of nearly 90% in relapsed or refractory (R/R) HL cases treated with nivolumab [Table S2] . In a similar fashion, the efficacy of pembrolizumab in brentuximab vedotin relapsed HL was examined, resulting in an overall response rate (ORR) of about 65%.
In one of the earliest studies exploring the role of FDG PET/CT in anti-PD1 immunotherapy, Dercle et al. found that among 16 patients with HL, 4 achieved complete metabolic response after 6 months of therapy. The same patients were then analyzed to determine patterns of response, and it was found that 9 out of 16 patients had objective response. These findings confirmed the reliability of FDG PET/CT in assessing response to immunotherapy . Metabolic PET parameters were also found to be significant outcome predictors. A study by Castello et al. found that metabolic PET parameters are significant outcome predictors in patients with R/R HL treated with pembrolizumab. A total of 43 patients were enrolled, and after a median follow-up of 19 months, the responder group had lower DS and SUV max values when compared with the non-responder group . Additionally, progression free survival (PFS) was found to be longer in the responder group. Despite the previously stated achievements of ICI in clinical settings, long-term benefits are still not adequately studied. Many recent studies have explored the potential benefits of combining immunotherapy with other treatment protocols for patients with advanced HL. One such study found that the combination of nivolumab with brentuximab vedotin plus doxorubicin and dacarbazine was highly effective, with more than 90% of patients achieving an ORR of 93%. These findings suggest that the combination of different treatment protocols may be a promising approach for improving outcomes in HL.
3.3.2. Brentuximab Vedotin (BV)
After the approval of immune checkpoint inhibitors for HL, another monoclonal antibody became available, BV, for the treatment of both adult and pediatric patients. The efficacy of the drug was explored in many trials through the use of FDG PET scans for therapy response . These studies showed an ORR of around 70%. Additionally, Kahraman et al. examined the efficacy of brentuximab vedotin in clinical settings through the use of FDG PET/CT scans to monitor therapy outcomes in cases of R/R HL . At the interim period and after a median follow-up of 16 months, PFS was significantly prolonged in patients with negative interim PET results compared to positive interim results
A recent study confirmed the previous observation that patients with negative iPET results have improved PFS and overall survival (OS) compared to those with positive iPET results.
Advanced HL cases were assessed with PET derived metrics to determine the predictive value. A study by Gavane et al. included 45 patients with R/R HL treated with brentuximab vedotin-based salvage therapy. It was observed that several baseline metabolic PET parameters, including metabolic tumor volume (MTV), total lesion glycolysis (TLG) and SUV peak, provide significant prognostic value in such patients . Previous research has established the role of BV in advanced cases of HL, but its efficacy in early-stage HL is not as well-known. In a study by Abramson et al., the use of combined AVD/BV without radiotherapy in 34 patients with non-bulky early-stage HL was explored. One cycle of BV was administered on days 1 and 15, followed by four cycles of AVD/BV. A complete response rate of 52% after the lead-in cycle of BV and 97% after two AVD/BV cycles was achieved, and the 3-year PFS rate was 94% [94]. In a study by Park and colleagues, the approach of 6 cycles BV consolidation therapy after 2-6 ABVD cycles in early-stage HL was explored [95]. A consolidation approach yielded a 95% complete response rate, and a three-year progression-free survival of 92%. Currently, there is a noticeable shift in emphasis toward incorporating BV as part of frontline therapy to observe therapy outcomes.
3.3.3. Chimeric Antigen Receptor Therapy (CAR-T)
Interestingly, CAR-T is the only immunotherapy that would require FDG PET/CT assessment during initial administration. In fact, two FDG PET/CT studies have to be carried out before CAR-T infusion. This involves performing FDG PET/CT at time of decision (TD), followed by second scan at time of transfusion (TT). Afterwards, another two scans will be performed to monitor therapy response at 1-month (M1) and 3-month (M3) intervals . This approach has attained high sensitivity and specificity of about 99% and 100% respectively. It is noteworthy that not all clinical centers adhere to this approach, as many clinicians rely on TT PET/CT as a baseline study. When assessing treatment response in clinical settings, multiple PET parameters are usually incorporated. These parameters are derived from values of 5 PS, SUVmax, and the variation between different time points (∆SUVmax), along with tumor volume analyses. Volumetric analyses rely on values of MTV. An unfavorable response is considered when there is less than 66% of SUVmax reduction between two time points . The successful results of BV therapy have helped to provide the infrastructure for CAR-T to implement CD30 as a potential target. In a study of 18 patients with R/R HL, Wang and colleagues found that treatment with anti-CD30 CAR T cells was feasible and tolerable. Patients in the study had received extensive prior treatment, including both conventional lymphodepletion regimens and more disease-controlling regimens. The ORR in the study was 39%, with 28% of patients showing stable disease at two months after therapy infusion. The median PFS was 6 months. Further support for the safety and efficacy of antiCD30 CAR-T therapy comes from a phase 1 trial conducted by Ramos et al. This trial included 9 patients with R/R HL or anaplastic large cell lymphoma. The study showed an ORR of 33%, demonstrating the feasibility and tolerability of this type of therapy [103]. A more recent trial enrolling 41 patients with R/R HL showed even more promising results, with an ORR of 72% and a one-year overall survival rate of 94%. This study suggests that anti-CD30 CAR-T therapy is a promising treatment option for patients with R/R HL . In a similar fashion, Voorhees et al. examined the predictive role of MTV prior to anti-CD30 therapy in HL. This study found that there was a strong association between PFS and MTV prior to lympho-depletion. Therefore, minimizing MTV value before CAR-T is found beneficial. The results from this study have broadened the field of research to include CAR-T. To date, there are 4 clinical trials underway to explore different potential uses of PET/CT for assessing therapy response in CAR-T patients.
3.4. FDG PET/CT in Non-Hodgkin’s Lymphoma (NHL)
Similarly, FDG PET/CT is of vital importance for outcome prediction and prognostication]. The only difference in NHL is that there are certain histologic subtypes that do not optimally express FDG avidity. Indolent NHL fall under the category of variable FDG avid NHL while aggressive NHL usually have moderate to high FDG avidity [Table 3]. Therefore, incorporation of FDG PET/CT is most acknowledged in response assessment of aggressive NHL.
3.4.1. Rituximab
Since the approval of FDG PET/CT by the FDA, a number of studies have been conducted to explore the efficacy of this treatment modality. Haioun et al. were among the first to examine the prognostic and predictive value of early FDG PET/CT imaging. In their study, 41% of all 90 patients received rituximab as part of the therapy protocol. These patients were then followed up to determine the prognostic outcome. It was concluded that patients with negative PET results had much more favorable outcomes, reflected by PFS and OS rates of 82% and 90%, respectively, as compared to 43% and 61% for those with positive PET results. More recently, a group of DLBCL patients treated with R-CHOP and evaluated by FDG PET/CT at the interim stage were prospectively enrolled in a study. The calculated 3-year PFS and OS rates in iPET negative patients achieved statistically significant superiority when compared to positive results. As of right now, metabolic PET parameters are being extensively studied in an attempt to outline their prognostic values. A recent study concerning baseline PET parameters in R-CHOP treated DLBCL patients was conducted. Metabolic PET parameters were used including SUVmax, SUVmean, MTV and TLG. These parameters were found to have prognostic value at baseline and interim intervals. In another study, tumor MTV values were found to be the most reliable parameters among all to determine survival outcome. This was recently shown by another study that confirmed the predictive value of baseline and interim MTV on survival outcome. It appears obvious by now that metabolic PET parameters can establish solid background for future response-adapted management approaches in NHL patients.
3.4.2. Immune Checkpoint Inhibitors (ICI)
Unlike HL, the results achieved in the previous literature using ICI in NHL are less encouraging. Despite having high safety profile, ICI failed to achieve optimal efficacy in NHL. The safety and efficacy of nivolumab in DLBCL were assessed in the previous single-arm phase II study by Ansell et al. . The study acknowledged suboptimal ORR despite the highly observed safety profile. On the other hand, Results from clinical trials of ICI combined with other immunochemotherapies appears more promising. Pembrolizumab was explored as a treatment for DLBCL in a study of 30 patients. This study found that the combination of pembrolizumab and R-CHOP resulted in an ORR of 90%, a CR of 77%, and a 2-year PFS of 83%. The findings of this trial indicate that combining the PD-L1 inhibitor atezolizumab with chemotherapy may be a promising treatment option for DLBCL. The combination of atezolizumab and R-CHOP (a type of chemotherapy) resulted in a high efficacy, with an ORR of 87.5% and durable responses in 80% of patients at 24 months. Based on previous research, it appears that combining immunotherapy with chemotherapy is more likely to result in favorable outcomes in terms of response and clinical outcome.
3.4.3. Chimeric Antigen Receptor Therapy (CAR-T)
A few studies have examined the value of FDG PET/CT in CAR-T, with mixed results. Shah et al. were among the first to examine MTV in a small group of NHL patients and found that nearly half the patients had non-measurable MTV values at M1 [Table S3]. These patients showed long-term remission over the following 2 years. The other half presented with measurable MTV and witnessed an early relapse.
It was concluded that SUVmax prior to therapy may help determine treatment eligibility and that DS and ∆SUVmax can help identify treatment failure . Recently, Galtier et al. conducted a multicentric cohort study which highlighted the high predictive values of both the 5 PS and MTV . This was also previously explored by Kuhnl et al., who found that Deauville criteria may predict the risk for CAR-T failure and help direct post-CAR-T management . Breen et al. have conducted more detailed analysis of SUVmax values at M1 and found that higher SUVmax values indicate higher risk for disease progression. SUVmax above 10 at M1 is regarded as a significant prognostic and predictive indicator in patients with stable disease or partial response . This was later confirmed by Al Zaki et al..In NHL, tumor burden was validated through the use of FDG PET/CT in a retrospective study by Wang et al. In fact, having high tumor burden at baseline was linked to more aggressive cytokine release syndrome . Bailly et al. enrolled a group of R/R NHL patients in order to demonstrate the added value of adequate disease control prior to therapy. Among all 40 patients, 33 cases were adequately managed prior to CAR-T. During TT PET/CT, adequately treated patients showed more favorable outcomes in terms of event free survival when compared to others . Moreover, 5 of the remainder 7 patients have witnessed early disease relapse . Therefore, it is essential to have adequate control prior to therapy initiation. Despite the encouraging outcomes of previous studies, more research is needed with larger cohorts to get a complete picture.
4.1. Monoclonal antibodies
It is interesting to note that the majority of these side effects are seen in the early stages of therapy. Most frequently, this pattern denotes immune system stimulation. Therefore, there is a high likelihood that these side effects might indicate a positive response to therapy.
4.1.1. Immunotherapy related inflammatory reactions
Immunotherapy related inflammation can affect any part of the body, as was seen in the ORIENT-1 trial. The purpose of this trial was to investigate the side effects of the antiPD1 ICI therapy called Sintilimab. Immunotherapy-related adverse events were primarily inflammatory in nature. The most common type of inflammatory event was enterocolitisrelated gastrointestinal inflammation, followed by pulmonary involvement and hepatitis. However, there have been no reports of ICI-related fatalities or significant morbidities in this trial, making it unjustifiable to discontinue therapy. Another study concerning ICI adverse events, conducted by Bajwa et al., found that the most commonly reported adverse effects were colitis, hepatitis, and myocarditi. In a study conducted by Petersen et al., FDG PET/CT scans were used to evaluate outcomes and detect related toxicities in patients receiving R-CHOP therapy . It was found that diffuse 18F-FDG uptake in the thyroid can indicate autoimmune thyroiditis and is linked to favorable outcomes . Additionally, another study reported that patients who experience imaging signs of at least one Immunotherapy related adverse event (most commonly colitis or arthritis) through PET/CT are more likely to have more favorable PFS than those who do not have any immunotherapy related adverse events . Therefore, it is advisable to document in the PET/CT report any and all side effects encountered during immunotherapy, even if they are not clinically relevant. In some cases, the results of a FDG PET/CT scan can be misinterpreted, leading to over-diagnosis or uncertainty about the disease process. To avoid this, it is advantageous to concentrate on follow-up and clinical context.
4.1.2. Reactive Changes
One potential indicator of immune system activity in the 18F-FDG PET/CT method of metabolic imaging is the inversion of the liver-to-spleen ratio (normally > 1. This may suggest immune activation prior to T cell proliferation, as well as reactive nodes in the primary tumor's drainage basin. This could potentially lead to misinterpretation during assessment. Diffuse reactive splenomegaly and bone marrow activity can occur following treatment with immunotherapy as well as chemotherapy. Usually, these findings do not result in serious consequences but it must be reported and monitored to ensure resolution after therapy discontinuation. Sarcoid-like reactive patterns have also been reported following ICI . FDG-avid bilateral symmetrical hilar and mediastinal lymphadenopathy can occur with or without clinical manifestation. Such patients are usually kept on follow-up to exclude metastatic processes.
4.1.3. Tumor Flare Reaction (TFR)
4.1.3. Tumor Flare Reaction (TFR)
In the last decade, TFR has been observed as a part of immunotherapy-related adverse effects in lymphoma. TFR is a clinical syndrome characterized primarily by diffuse FDG-avid generalized lymphadenopathy and splenomegaly, along with other clinical manifestations. It bears a striking resemblance to hyperprogression. The only observed difference between hyperprogression and TFR in lymphoma is the underlying cause. TFR is an immune-mediated process, not a disease progression. This difference can be suspected through FDG PET/CT and confirmed by biopsy. In fact, TFR is more of a clinical manifestation of the pseudoprogression pattern that is sometimes seen during the initial therapy response. Therefore, its incidence should not be used as a reason to discontinue treatment. A PET/CT examination of the patient showed increased metabolic activity of enlarged lymph nodes after R-CHOP treatment and allogeneic transplantation [140]. However, a biopsy of the lymph node revealed extensive reactive T cell infiltration, with no signs of lymphoma cells [140]. Re-examination of PET/CT scans showed no obvious enlargement or increased metabolic activity of lymph nodes after 3 months. More recently, another case report was published describing the clinical and pathological manifestations of TFR . The patient's TFR was evident after 4 cycles of ICI NHL treatment, as FDG PET/CT showed enlarged and progressing lymphadenopathy above the diaphragm . This was accompanied by other clinical manifestations such as fever, skin rash, joint pain, and poor appetite. After further investigation, TFR was suspected . A left axillary lymph node biopsy was done, ruling out lymphomatous involvement. The patient's clinical condition improved after the glucocorticoid intake therapy was continued, and the follow-up FDG PET/CT was negative thereafter.
4.2. Chimeric Antigen Receptor Therapy (CAR-T)
CAR-T cell therapy has been found to cause more immediate and severe side effects than monoclonal antibodies. In order to ensure the best possible outcomes, it is essential to investigate, study, and document these side effects.
4.2.1. Cytokine release syndrome (CRS)
CRS is the most common side effect of CAR-T therapy. This syndrome may start on the first day after the therapy transfusion and last for up to 9 days. Although FDG PET/CT does not have a direct role in diagnosing CRS, research has indicated that metrics derived from PET can predict the occurrence of CRS. Studies investigating CAR-T for R/R NHL have shown that the tumor burden is strongly correlated with the severity of CRS. Additionally, recent research has shown that a high tumor burden, as reflected by SUV average (SUVAvg), MTV, and TLG, is a significant risk factor for developing any grade of CRS.
4.2.2. Immune Effector Cell Associated Neurotoxicity Syndrome (ICANS)
The clinical manifestations of ICANS include encephalopathy, delirium, and altered mental status . Unlike CRS, the role of FDG PET/CT in ICANS is more widely acknowledged. Brain FDG PET/CT can help exclude differential diagnoses of these cases. In previous studies, neurological deficits associated with ICANS have been observed using brain FDG PET/CT. These deficits typically manifest as areas of reduced metabolism on Brain FDG PET/CT scans. This was demonstrated by Rubin et al., who found that these deficits were associated with CAR-T cell toxicity up to 2 months after transfusion. In a cohort of six patients, five showed EEG abnormalities that corresponded to hypometabolic areas on PET/CT scans . More recently, Paccagnella et al. described a case of ICANS in a male patient who developed symptoms 4 days after CAR-T infusion . FDG PET/CT of the brain showed global hypometabolism, particularly in the left hemisphere and frontal regions. Clinical manifestations such as ideo-motor slowing, nominal aphasia, and myoclonic tremor was associated with this condition. However, despite the diffuse slowing seen on EEG, the MRI showed no pathological involvement. After intravenous steroid administration, the patient in the study responded completely. The findings of the current study are consistent with those of a recent case report by Vernier et al., in a patient with DLCBL . PET/CT brain scans performed before and after treatment showed bilateral and diffuse hypometabolism in the parietal and temporal cortex, with no abnormal findings on MRI.
5. Current Challenges and Future Prospects
Many challenges in the field of lymphoma immunotherapy have been overcome through the creation of optimal future plans. This had led to the creation of a new therapyspecific PET criteria, as well as the exploration of PET-derived metrics for therapy response and outcome prediction.
5.1. Current Challenges
5.1.1. Limitations in Low-income and conflict region
The lack of access to optimal cancer diagnostic services is a major problem in many parts of the world, especially in conflict-affected areas and low-income countries . This can lead to sub-optimal care and a lack of uniformity in cancer diagnosis and treatment. Establishing international recovery programs is necessary to overcome these difficulties.
5.1.2. Financial burden
Since the FDA approval of ICI and CAR-T therapies, many barriers and considerations have arisen in regards to their widespread use. While some countries are progressing with these therapies, their implementation remains uneven among different nations. To date, many countries are incapable of providing this novel therapies due to lack of financial support, unavailability or financial instability . This is especially apparent in low-income countries and conflict regions. Under these circumstances, medical specialists must carefully select cost-effective treatments for patients and engage them in decision-making to ensure they understand the potential financial consequences of their options.
5.1.3 Volumetric PET parameters: promising yet overlooked
To date, most institutions rely on semi-quantitative PET parameters through the use of SUVmax values . This is because to SUVmax familiarity among interpreter physicians, validity in medical literature, and appropriate operator reproducibility. However, SUVmax is sensitive to image noise and motion, and its value is dependent on image quality . It should be noted that PET-CT scanners with high spatial resolution tend to produce images with high SUVmax values . Therefore, direct comparison between images produced by different scanners may not be possible . Another factor to consider when interpreting SUV values is that they can be affected by blood glucose levels and uptake time (i.e., the time interval between injection and scanning. On the other hand, volumetric parameters such as MTV and TLG have several advantages over SUV, including being less noise-sensitive. Earlier studies have found that using a variety of PET parameters, including both volumetric and semiquantitative parameters, is an efficient strategy. MTV and TLG have been found to be connected with a higher risk of several different types of cancer. Additionally, previous studies found evidence that volumetric indices can be used to predict prognosis. However, some institutions lack access to these parameters due to limitations in software and experiences. Another issue is the lack of consensus in determining the tumor boundary, which is necessary for reproducibility. Nonetheless, implementing these parameters in future practices can provide additional usefulness once the current barriers are overcome.
5.1.4. Lack of harmonization in implementing Novel response criteria
Tumor response criteria have been modified over the years to accommodate the recent advancements in lymphoma therapy. Different patterns of response to immunotherapy in lymphoma patients as seen on PET/CT scans necessitated the development of new, therapy-specific criteria . However, not all institutions have adopted these new criteria. Many instead are still relying on the Lugano criteria which does not take into account pseudoprogression . There is therefore a need to focus on harmonizing response assessment in order to optimize response evaluation.
5.2. Future Prospects
5.2.1. FDG Alternatives
The current focus in lymphoma treatment is on providing targeted therapy using theranostics. The CXC chemokine receptor 4 (CXCR4) is a promising cancer treatment target. This transmembrane receptor is involved in hematopoietic stem cell migration and homing to the bone marrow . Multiple studies have demonstrated the imaging and targeting capabilities of 68Ga-Pentixafor for CXCR4-expressing lymphomas. Fibroblast activation protein inhibitor (FAPI) is currently being studied for its potential use as both a diagnostic and therapeutic tool in lymphoma. Preliminary results are encouraging in terms of safety and efficacy. FAP is overexpressed by cancer-associated fibroblasts present in the tumor microenvironment, which leads to high tumor uptake and very low accumulation in normal tissues, achieving excellent results . Incorporation of 18 Fluorine and Fludarabine (a chemotherapy drug already used in low-grade lymphomas) was examined in variable and low FDG-avid lymphomas. Studies in animals and humans have demonstrated lower uptake of these drugs in inflammatory cells compared to FDG and a better correlation with histology than the latter. The sensitivity of18 F-Fludarabine for detecting indolent lymphoma lesions has been found to be good, with reports indicating that it could potentially be used to replace FDG PET/CT imaging for such cases.
5.2.2 Radiomics and machine learning
The use of FDG PET/CT for predictive and prognostic purposes in lymphoma has been well established over the past few decades. However, many other features of this imaging modality have only recently been explored. Machine learning algorithms can be used to detect and measure overall tumor burden and are currently being examined. Frood et al. investigate the potential benefits of using a radiomic model to indicate 2-year PFS of HL patients . The study found that the model was useful, but that more research is needed to confirm efficacy. Radiomics can be used to minimize time and effort during PET/CT assessment. Radiomic-based segmentation techniques studied by Sollini et al. can serve as a useful tool in clinical practice for assessing involved lesions in HL patients, with an overall accuracy of 82% . This radiomic technique is based on the utilization of lesion similarity analysis when assessing involved lesions in HL patients. These methods have shown to be useful in histological prediction, prognostic evaluation, and the definition of bone marrow involvement. Nonetheless, more clinical research is necessary to determine the efficacy of these AI algorithms
6. Conclusions
Cancer immunotherapy continues to deliver alternative and superior therapy choices for certain stages and subtypes of lymphoma.
Advancement in this field necessitates the development of optimal response criteria that can provide predictive and prognostic information through the use of metabolic parameters.
FDG PET/CT plays a key role in the newly developed response criteria and several FDG PET/CT-based criteria have been proposed to address all patterns of response to therapy.
Including several response types such as indeterminate response, pseudoprogression and hyperprogression using several metrics, such as SUV, MTV and TLG.
Certainly, more harmonization and consensus are still desired, and the future holds promise for superior immunotherapies and more optimized criteria for assessing response and impacting patient outcomes.
References
They have been well selected, recently published and enough in number
Author Response
Thank you for your insightful comments. It is very motivating to know that our work is appreciated and will hopefully have a positive impact on such an important topic.
